# Nanowarming and ice-free cryopreservation of large sized, intact porcine articular cartilage

Peng Chen [1], Shangping Wang[1], Zhenzhen Chen[2], Pengling Ren[1,3], R. Glenn Hepfer[1,4], Elizabeth D. Greene[2], Lia H. Campbell[2], Kristi L. Helke [5], Xingju Nie [6], Jens H. Jensen [6], Cherice Hill[1,4], Yongren Wu[1,3], Kelvin G. M. Brockbank[1,2] & Hai Yao [1,3,4 ✉]

Successful organ or tissue long-term preservation would revolutionize biomedicine. Cartilage cryopreservation enables prolonged shelf life of articular cartilage, posing the prospect to broaden the implementation of promising osteochondral allograft (OCA) transplantation for cartilage repair. However, cryopreserved large sized cartilage cannot be successfully warmed with the conventional convection warming approach due to its limited warming rate, blocking its clinical potential. Here, we develop a nanowarming and ice-free cryopreservation method for large sized, intact articular cartilage preservation. Our method achieves a heating rate of 76.8 °C min$^{-1}$, over one order of magnitude higher than convection warming (4.8 °C min$^{-1}$). Using systematic cell and tissue level tests, we demonstrate the superior performance of our method in preserving large cartilage. A depth-dependent preservation manner is also observed and recapitulated through magnetic resonance imaging and computational modeling. Finally, we show that the delivery of nanoparticles to the OCA bone side could be a feasible direction for further optimization of our method. This study pioneers the application of nanowarming and ice-free cryopreservation for large articular cartilage and provides valuable insights for future technique development, paving the way for clinical applications of cryopreserved cartilage.

[1] Department of Bioengineering, Clemson University, Clemson, SC, USA. [2] Tissue Testing Technology LLC, North Charleston, SC, USA. [3] Department of Orthopaedics, Medical University of South Carolina, Charleston, SC, USA. [4] Department of Oral Health Sciences, Medical University of South Carolina, Charleston, SC, USA. [5] Department of Comparative Medicine, Medical University of South Carolina, Charleston, SC, USA. [6] Department of Neuroscience, Medical University of South Carolina, Charleston, SC, USA. ✉email: haiyao@clemson.edu

Organ or bulk tissue transplantation has saved millions of lives and improved the quality of life for patients suffering from organ failure or tissue diseases[1,2]. However, the number of organs and tissues ready for transplantation is far from reaching the patient population's needs. For instance, in 2020, 39,036 organ transplantations were performed in the United States while 55,121 new candidates were added to the waitlist[3]; many patients die while waiting for organ transplantation. Despite the huge unmet demand for organs and tissues, the disappointing fact is that many organs and tissues are not viable for implantation and are discarded because they reach the maximum postmortem preservation time limit[1,4]. In the case of organ transplantation, up to 70% of donor organs expire before reaching patients[1,4]. Achieving long-term organ and tissue preservation is critical to improving utilization rates and alleviating existing shortages.

Articular cartilage defects are common knee problems, affecting 60–66% of patients undergoing arthroscopic procedures[5–8]. Due to the limited regenerative capacity of articular cartilage, cartilage defects can develop into osteoarthritis (OA) without proper surgical intervention. OA affects more than 25.6 million adults and results in nearly $200 billion in related costs every year in the United States[9]. As the population grows and ages, the number of OA patients is projected to increase dramatically[10]. Currently, there is no cure for OA, demonstrating the need for continued research in this area. Osteochondral allograft (OCA) transplantation is an appealing cartilage repair strategy to manage OA progression, particularly for young and physically active patients with large defects[11]. Compared to other treatment options such as microfracture, autologous chondrocyte implantation, osteochondral autograft transplantation, and emerging tissue engineering approaches, OCA transplantation immediately replaces the defect site with functional full-thickness hyaline cartilage, requires only a single surgical procedure, and avoids the issue of donor site mortality[12,13]. OCA transplantation is also a preferred surgical choice for patients with widespread cartilage damage, subchondral bone injury, or failed prior cartilage surgery[12,13]. The mean 10-year survival rate of OCA transplantation is 78.7%[14], and 75–88% of patients are able to return to sports after transplantation[15]. Clinical use of OCA transplantation for cartilage restoration has increased in popularity over the past decade[16], and a greater need for OCAs is expected in the future. However, the availability of fresh OCAs is limited. Current gold standard hypothermic storage methods for OCA storage can only preserve OCA functional properties for up to 28 days before implantation[17,18]. This time frame is often insufficient considering the extended time needed for surgical planning including OCA harvest, transportation, a minimum 14-day infectious disease screening, graft matching, patient preparation, and other related logistics. Nearly 30% of OCAs are discarded without implantation in the United States[19], further challenging OCA availability and restricting the broad implementation of OCA transplantation.

Long-term preservation methods have been utilized to improve the shelf life of OCAs. Early attempts proposed to use of a simple freezing method for OCA preservation, but later studies revealed that cartilage preserved through this method has low cell viability and poor extracellular integrity, leading to compromised outcomes in both animal and human transplantation studies[20,21]. Vitrification, a process that transfers a liquid solution into a glassy solid without ice formation, was then introduced. In this approach, OCAs are immersed in high-concentration cryoprotectant agent (CPA) solutions and then cooled down to deep subzero temperatures. The incorporation of CPAs significantly reduces the cooling rate requirements, making it feasible to achieve ice-free storage with routine lab equipment. Vitrification

has been successfully used to cryopreserve OCAs in small volumes in multiple species, including rabbits[22,23], pigs[24,25], and humans[26,27]. In these studies, the size of articular cartilage specimens was limited up to 10 mm in diameter, and the volume of surrounding CPA solution was kept under 3 ml. Since 36–54% of cartilage defects are larger than 1 cm$^2$ [5,7], cryopreservation of large OCAs is needed but remains challenging, particularly with regard to the warming process[28,29]. Currently, vitrified samples are commonly warmed through a convection strategy using a warm water bath (37 °C). The heat is gradually transferred from the water bath, through the surrounding CPA, and finally into the tissue. However, due to the heat transfer limit, convection warming could fail to achieve the CPA critical warming rates (normally folders or orders of magnitude higher than the critical cooling rates) in larger samples, causing detrimental ice formation. In addition, nonuniform heating during convection warming could cause thermal stress exceeding the sample's mechanical compliance, leading to cracking and damage of the preserved samples[30–32].

Several volumetric warming methods have been proposed to resolve the warming rate and uniformity issues, including laser warming and microwave warming. The laser warming method quickly heats the sample in the presence of laser-absorbing molecules, such as gold nanorods[33,34]. Despite its ultra-rapid warming efficiency, its application is currently limited to the warming of microliter samples, such as cells and embryos, owing to the restriction of laser penetration depth[33,34]. Microwave warming can directly warm the sample by exciting its molecule electrical dipoles with high frequency (hundreds of MHz to GHz) electromagnetic fields[35,36]. Although this method can achieve a high warming rate, it is prone to thermal runaway issues, a phenomenon of increased temperature differences between the warmed site and cold site due to the temperature-dependent dielectric property of the heated sample, causing unfavorable hot spots[28,35,36]. The heating efficiency or profile of microwave warming is also dependent on the sample shape[35,37], further complicating its general applications.

Recently, a nanowarming technique has been developed for large sample cryopreservation[28,30–32,38–41]. Vitrified samples loaded with magnetic iron oxide nanoparticles (mIONPs) are exposed to a low radiofrequency alternative magnetic field (hundreds of kHz) for warming, as opposed to traditional convection warming that relies on passive heat transfer from the water bath to the samples. Unlike the high-frequency microwave warming method[37,42], the low radiofrequency magnetic field itself cannot quickly warm the tissues; however, it can efficiently induce heat through the mIONPs, where the magnitude and frequency of the magnetic field and the concentration of the mIONPs can be tuned to achieve the desired heating rate[30]. As the low radiofrequency magnetic field can penetrate the tissue without attenuation, uniform heating can be accomplished once the magnetic field is homogeneous and the mIONPs are evenly distributed within the tissue. Nanowarming combined with vitrification has been applied to preserve porcine arteries[30], porcine heart valves[30], rat kidneys[32], and rat hearts[31,38] with varying degrees of success. Nanowarming for organ or tissue preservation is still in its infancy and prior focus has been limited to thin or highly vascularized biological systems. Thin tissues like heart valve leaflets and arteries enable fast heat transfer with nanoparticles in close proximity, while vascularized systems (e.g., rat hearts and rat kidneys) facilitate nanoparticle distribution throughout the organ. Additionally, it is noteworthy that the actual volume of organs warmed using nanowarming is small; the volume of the rat heart and kidney, for instance, are both below 1 ml[31,32,38]. No attempts have been made into the application of

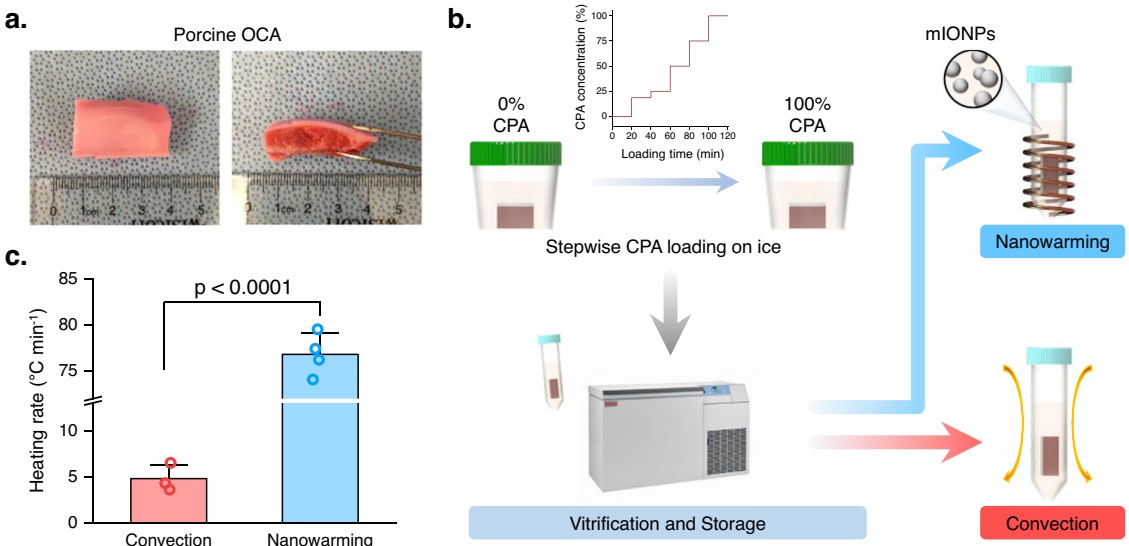

**Fig. 1 Large OCA nanowarming and ice-free cryopreservation vs. conventional cryopreservation. a** Porcine femoral trochlea OCA used in this study. The OCA size is ~30 mm × 20 mm × 14 mm (length × width × thickness). **b** Schematic flow of two cryopreservation methods. For both strategies, the porcine OCA was first loaded with CPA solution at 4 °C following a stepwise CPA loading protocol. The OCA was immersed in CPA solution with increasing concentration (0%, 18.5%, 25%, 50%, 75%, and 100%) sequentially for 20 min in each step. Then the OCA was vitrified and stored in a mechanical freezer. mIONPs were added immediately after the final step of 100% CPA loading at a concentration of 2 mg ml$^{-1}$ for the nanowarming group. During rewarming, the nanowarming of vitrified OCAs was performed in a customized coil generating a radiofrequency alternative magnetic field while convection warming was conducted with the vitrified OCAs in a warm water bath (37 °C). After rewarming, all OCAs went through an unloading step in a reverse manner as the loading protocol to remove the CPA. **c** Heating rate achieved through convection warming and nanowarming (Supplementary Fig. 2d, e). $n = 3$ independent samples for the convection group and $n = 4$ independent samples for nanowarming group. $p$-value was determined with a two-sided $t$-test. All data depict mean ± standard deviation.

the nanowarming technique for the preservation of large, dense, and avascular cartilage tissues.

Here, we developed a nanowarming and ice-free cryopreservation method by integrating cartilage vitrification and nanowarming techniques for the preservation of large OCAs. Using this method, the heating rate during the warming process was improved over an order of magnitude, overcoming the difficulty of slow rewarming with convection in a warm bath. A systematic characterization of cartilage cell and tissue level behaviors was also performed to provide a wide spectrum assessment of our method for large OCA preservation. Our results showed that nanowarming and ice-free cryopreservation outperformed the traditional convection method in combination with ice-free cryopreservation by rescuing more live cells and maintaining a higher level of cell metabolic activity. Compared to convection warming, our method also displayed a trend of better performance in maintaining cartilage material and mechanical properties. Along with computational modeling and magnetic resonance imaging (MRI), the experimental observation of depth-dependent preservation manner was recapitulated and the delivery of mIONPs to the subchondral bone was hinted at as a possible means to further optimize the performance of our method for large OCA preservation. This study pioneers the application of ice-free vitrification and nanowarming for large articular cartilage preservation and provides valuable insights for future technique development and the translation of this technique to the clinical practice of OCA transplantation.

## Results

**Nanowarming and ice-free cryopreservation enables long-term large OCA preservation.** Large OCA samples (length × width × thickness, ~30 mm × 20 mm × 14 mm) were harvested (Fig. 1a) and loaded with CPA solutions at 4 °C through a stepwise protocol to ensure vitrification during the cooling process (Fig. 1b).

Successful OCA vitrification was achieved without any cracks or ice formation observed in the sample (Supplementary Fig. 1a). The cooling rate also met the vitrification requirements (Supplementary Fig. 1b, c). In the nanowarming group, mIONPs were added right after the final step of CPA loading. Multiple CPA solutions, including VS55, VS70, and VS83, were tested in this study. After vitrification and storage, the vitrified OCA was then warmed through one of two mechanisms: convection warming or nanowarming. Convection warming was achieved by placing the sample in a 37 °C water bath while nanowarming was implemented in a customized coil with a radiofrequency alternating magnetic field (Fig. 1b). Temperature measurement in pure CPA solutions with 2 mg ml$^{-1}$ mIONPs showed that the heating rate was 76.80 ± 2.27 °C min$^{-1}$ for the nanowarming method, which was over one order magnitude higher than the 4.80 ± 1.51 °C min$^{-1}$ achieved by convection warming (Fig. 1c and Supplementary Fig. 2). During the rewarming process, no cracks were observed in the nanowarming group while cracks were clearly seen in the surrounding CPA in the convection group. These results, in agreement with prior literature[28,30–32,38–41], demonstrate the capacity for rapid sample rewarming using nanowarming. Therefore, our method enables the long-term preservation of large OCAs.

**Nanowarming and ice-free cryopreservation retain cell functions.** An alamarBlue assay was applied to evaluate cartilage cell metabolic activity after rewarming. In alignment with our previous study[25], the cartilage preserved with VS83 displayed a much higher metabolic activity level than the cartilage preserved with VS70 in both convection and nanowarming groups (Fig. 2a). Results with the most commonly used VS55 displayed the lowest level of cell metabolic activity (Supplementary Fig. 3). Therefore, VS83 was used for all subsequent studies. In addition, the cartilage in the nanowarming group had significantly higher cell

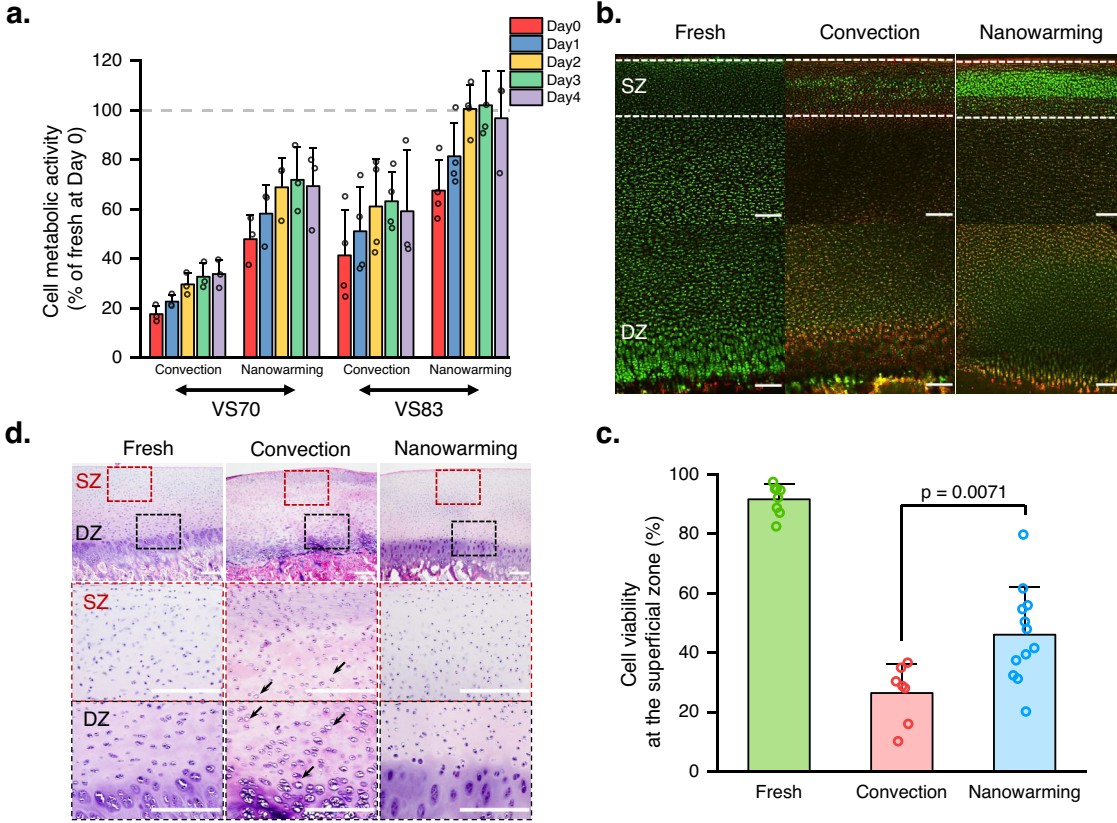

**Fig. 2 Nanowarming outperforms convection warming in retaining cell function for large OCA preservation. a** Cell metabolic activity results were measured by alamarBlue assay. The results were normalized to the value of fresh control cartilage measured at day 0 of each batch experiment. The gray dashed line indicates 100% recovery, reaching the level of cell metabolic activity of fresh cartilage ($n = 3$ independent samples for both convection and nanowarming groups with VS70 solutions. $n = 4$ independent samples for both convection and nanowarming groups with VS83 solutions). **b** Representative fluorescence live/dead staining results measured with samples collected immediately after rewarming. SZ superficial zone, DZ deep zone. The white dashed line indicates the superficial region for the quantification analysis of cell viability shown in (**c**). Scale bars represent 200 μm. **c** Cell viability quantified at the superficial layer of fluorescence live/dead staining images in fresh ($n = 8$ independent samples), convection ($n = 7$ independent samples), and nanowarming ($n = 12$ independent samples) cartilages. *p*-value was determined with one-way ANOVA with Bonferroni post-hoc test. **d** Representative Hematoxylin & Eosin (H&E) staining images of fresh, convection, and nanowarming cartilages were collected immediately after rewarming. SZ superficial zone, enlarged image shown in the red dashed box; DZ deep zone, enlarged image shown in the black dashed box. Scale bars represent 200 μm. The black arrows highlight locations with white space, indicating possible ice formation. All data depict mean ± standard deviation. Note: mIONPs in the nanowaming group are loaded through the immersion approach for all data presented here.

metabolic activity than cartilage warmed with convection (Fig. 2a). In particular, cartilage nanowarmed with VS83 showed full cell metabolic activity recovery after 2-day culture in media (Fig. 2a). These results, along with the cooling and heating rate measurements (Supplementary Figs. 1, 2), indicate better preservation of cell function with nanowarming compared to convection warming. Cartilage cell viability was examined employing fluorescence live/dead staining immediately after rewarming. Fresh cartilage cells were alive throughout the sample, exhibiting strong green fluorescence signals (Fig. 2b). In the convection group, only a small portion of cells in the cartilage superficial zone was alive while cartilage in the nanowarming group had more live cells in both the superficial and deep zones (Fig. 2b). Quantitative analysis of cell viability at the superficial zone showed that nanowarmed cartilage had significantly higher cell viability than convection warmed cartilage (Fig. 2c). Hematoxylin & Eosin (H&E) staining also revealed that cartilage in the convection group had more white space than the cartilage in both fresh and nanowarming groups (Fig. 2d). These white empty spaces are believed to be a sign of ice formation during preservation[22], suggesting the failure of cartilage rewarming by convection. Safranin O staining was also performed and showed

more white space in convection-warmed cartilage than fresh and nanowarmed cartilage (Supplementary Fig. 4). However, similar color appearances were found among the three groups (Supplementary Fig. 4), indicating minimal if any glycosaminoglycan content alterations after both convection warming and nanowarming.

**Nanowarming and ice-free cryopreservation trends toward better tissue-level outcomes**. Articular cartilage is mainly composed of water, collagen, and proteoglycans, exhibiting electromechanical properties. Alterations of tissue material properties can therefore adversely affect cartilage function. Currently, chondrocyte viability is the major parameter to evaluate the outcome of different articular cartilage cryopreservation strategies; tissue matrix properties are largely overlooked. In this study, electrical conductivity measurements with cartilage equilibrated in isotonic and hypotonic KCl solutions were used to assess changes in cartilage conductivity and fixed charge density (FCD) following convection warming and nanowarming processes (Fig. 3a)[43,44]. FCD is a measure of the negative charges attached to the proteoglycans[43,44]. Results showed that cartilage

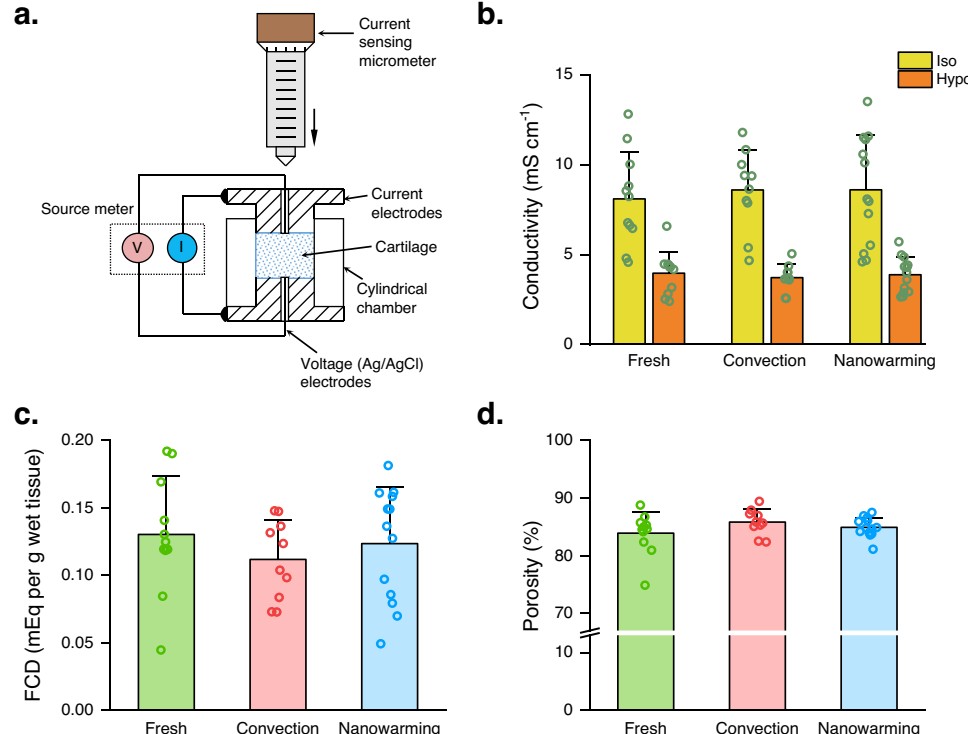

**Fig. 3 Nanowarmed cartilage is more alike fresh cartilage regarding cartilage electrical and compositional properties. a** Schematic of electrical conductivity measurement. Cartilage electrical conductivity and FCD were determined with a customized conductivity apparatus consisting of a current sensing micrometer for cartilage thickness measurement, two current electrodes, two Ag/AgCl voltage electrodes, and a cylindrical chamber. **b** Electrical conductivity of fresh ($n = 11$ independent samples), convection ($n = 10$ independent samples), and nanowarming ($n = 13$ independent samples) cartilage measured under both isotonic and hypotonic conditions. **c** FCD of fresh ($n = 11$ independent samples), convection ($n = 10$ independent samples), and nanowarming ($n = 13$ independent samples) cartilage. **d** Porosity of fresh ($n = 11$ independent samples), convection ($n = 10$ independent samples), and nanowarming ($n = 13$ independent samples) cartilage. All data depict mean ± standard deviation. Note: mIONPs in the nanowaming group are loaded through the immersion approach for all data presented here.

conductivities were similar among the fresh, convection, and nanowarming groups for both the isotonic and hypotonic conditions (Fig. 3b). Although no significant differences in FCD content were found among the three groups, the convection group had a lower FCD value than the fresh and nanowarming groups (Fig. 3c). Tissue porosity was also similar among the three groups (Fig. 3d). These results indicate that neither convection nor nanowarming had major impacts on cartilage electrical and compositional properties but nanowarmed cartilage was more similar to fresh cartilage than convection warmed cartilage.

Cartilage biomechanical properties were assessed to evaluate the impact of both warming strategies on tissue mechanical function. Microindentation tests were performed on intact osteochondral plugs punched from the large OCAs for all three groups (Fig. 4a). The indentation results showed no statistically significant differences among the groups (Fig. 4b, c). However, the cartilage equilibrium contact modulus, a parameter characterizing cartilage matrix mechanical strength, was smaller in the convection group than the fresh and nanowarming groups (Fig. 4b). Permeability was also higher in the convection group than the other two groups (Fig. 4c). These results further indicated better tissue level performance with the nanowarming method compared to the convection method. Since the microindentation test mainly explores the cartilage mechanical response at the surface layer (~200 μm), a confined compression experiment was conducted to investigate the full-thickness cartilage mechanical properties (Fig. 4d). The bone side of the osteochondral plugs was trimmed off, and cylindrical articular cartilage samples were placed into the confined chamber for the

relaxation tests. Confined compression experiments showed that cartilage from both convection and nanowarming groups displayed significantly lower aggregate moduli than the fresh cartilage group while no significant differences in permeability were found (Fig. 4e, f). These results indicate that the cartilage surface layer was better preserved than the middle or deep regions, which echoes the fluorescence live/dead staining results (Fig. 2b, c) and demonstrates a depth dependency for cartilage preservation.

**Computational modeling reveals depth-dependent manner of large OCA nanowarming preservation.** Cell and tissue level examinations demonstrate improved preservation of large articular cartilage with nanowarming compared to convection warming, particularly in the superficial layer. To further study nanowarming preservation capacity, computational modeling was used to simulate the cartilage nanowarming process. As there is very limited VS83 thermal data in the literature, we compared thermal behaviors between the VS83 solutions and the VS55 solutions. Direct measurements of solution density revealed that both VS83 and VS55 have similar solution densities of approximately 1100 kg m[-3] (Supplementary Fig. 5a). Under identical conditions, temperature measurements from cooling, convection warming, and nanowarming all showed very similar temperature profiles between VS55 and VS83 (Supplementary Fig. 5b–f), indicating that the thermal properties of VS55 and VS83 are alike. Therefore, the thermal parameters of VS55 solutions were used in the later modeling study. As a major

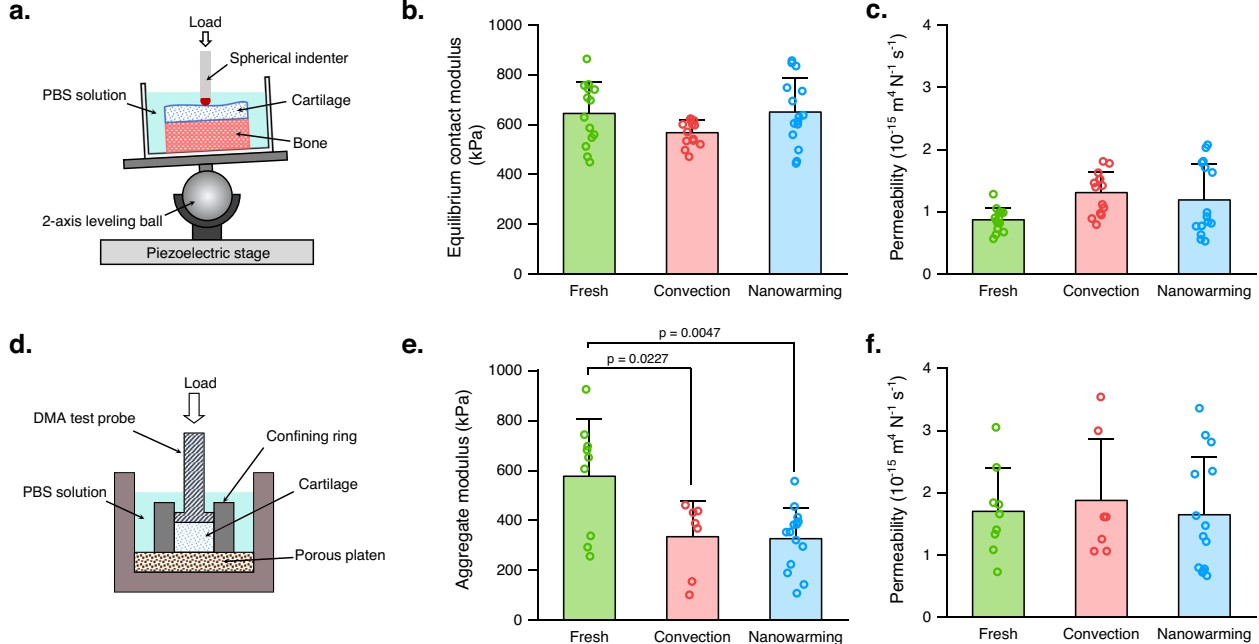

**Fig. 4 Depth-dependent changes in cartilage mechanical properties following nanowarming. a** Schematic of microindentation experiments. Cartilage surface mechanical properties were measured using a microindentation system consisting of a piezoelectrical stage, a 2-axis leveling ball, a sample chamber, and a spherical ruby ball indenter (1 mm in diameter). Osteochondral plugs were tested while immersed in PBS solutions. **b** Equilibrium contact modulus results of fresh ($n = 14$ measurements from 5 independent samples), convection ($n = 14$ measurements from 5 independent samples), and nanowarming ($n = 15$ measurements from 5 independent samples) cartilage determined through microindentation tests. **c** Permeability results of fresh ($n = 14$ measurements from 5 independent samples), convection ($n = 14$ measurements from 5 independent samples), and nanowarming ($n = 15$ measurements from 5 independent samples) cartilage determined through microindentation tests. **d** Schematic of a confined compression test. DMA dynamic mechanical analyzer. Cartilage plugs were compressed between a test probe (5 mm in diameter) and a porous platen while sitting in a confining ring. **e** Aggregate modulus results of fresh ($n = 9$ independent samples), convection ($n = 7$ independent samples), and nanowarming ($n = 14$ independent samples) cartilage measured by confined compression tests. *p*-value was determined with one-way ANOVA with Bonferroni post-hoc test. **f** Permeability results of fresh ($n = 9$ independent samples), convection ($n = 7$ independent samples), and nanowarming ($n = 14$ independent samples) cartilage measured by confined compression tests. All data depict mean ± standard deviation. Note: mIONPs in the nanowaming group are loaded through the immersion approach for all data presented here.

model input, the heat generation rate induced by nanowarming was also determined. Modeling results with VS83 containing mIONPs revealed a linear relationship between the heating rate and heat generation rate ($q$), supporting the uniform heating feature of the nanowarming procedure (Supplementary Fig. 5g, h). Taking the heating rate of 76.80 °C min$^{-1}$, measured in VS83 with 2 mg ml$^{-1}$ mIONPs, we obtained the corresponding value of the heat generation rate, $2.956 \times 10^6$ W m$^{-3}$ (Supplementary Fig. 5h). To further validate the value of the heat generation rate determined above, the heating rate of VS83 with 1 mg ml$^{-1}$ mIONPs during nanowarming was measured and compared to the modeling value. Our results showed that the experimental heating rate with 1 mg ml$^{-1}$ mIONPs was 41.39 ± 1.21 °C min$^{-1}$, which matched the modeling value of 38.40 °C min$^{-1}$ (7.8% relative error) (Supplementary Fig. 5h). Therefore, the value of the heating generation rate for the VS83 with 2 mg m$^{-1}$ mIONPs was estimated to be $2.956 \times 10^6$ W m$^{-3}$ and used in the subsequent models. Then, mIONP distribution was examined through visual inspection and magnetic resonance imaging (MRI) scans of OCAs. It was found that the OCA appearance and its MRI contrast did not change with increased OCA incubation time in mIONP solutions (Fig. 5a, b and Supplementary Fig. 6), indicating that mIONPs were unable to penetrate both the cartilage and bone portion of the OCAs with the current immersion loading protocol. Therefore, in the computational model consisting of an OCA and CPA solution in a cylindrical tube (Fig. 5c), no heat source was defined in the OCA region.

Modeling results demonstrated nonuniform temperature distribution within the OCA (Fig. 5d), which was expected since no mIONPs were present inside the OCA. For comparison, the convection warming process was also modeled. Results showed a much slower OCA temperature rise through convection warming (Fig. 5e). By characterizing the temperature profile at different cartilage depths, a depth-dependent temperature profile was observed in the nanowarming modeling, and the temperature rise was much faster in nanowarming than in convection warming (Fig. 5d–f). The heating rate was the highest at the cartilage surface and gradually decreased with depth (Fig. 5g). Experimental temperature data were also collected at the cartilage surface to validate the modeling findings. The results showed that the nanowarming group presented a much faster temperature rise and significantly higher heating rate than the convection warming group (Fig. 5f, g). The modeling results matched the experimental observations (Fig. 2b, c and Fig. 4), demonstrating the benefit of nanowarming for preserving the cartilage surface layer and revealing the limited capacity of nanowarming for preserving the full-thickness cartilage.

**Bone-side nanoparticle delivery augments the performance of nanowarming large OCA preservation.** Protocol optimization could further improve the performance of our method for the preservation of large OCAs. Owing to the absence of mIONPs inside the OCAs, our method heated up the surrounding CPA

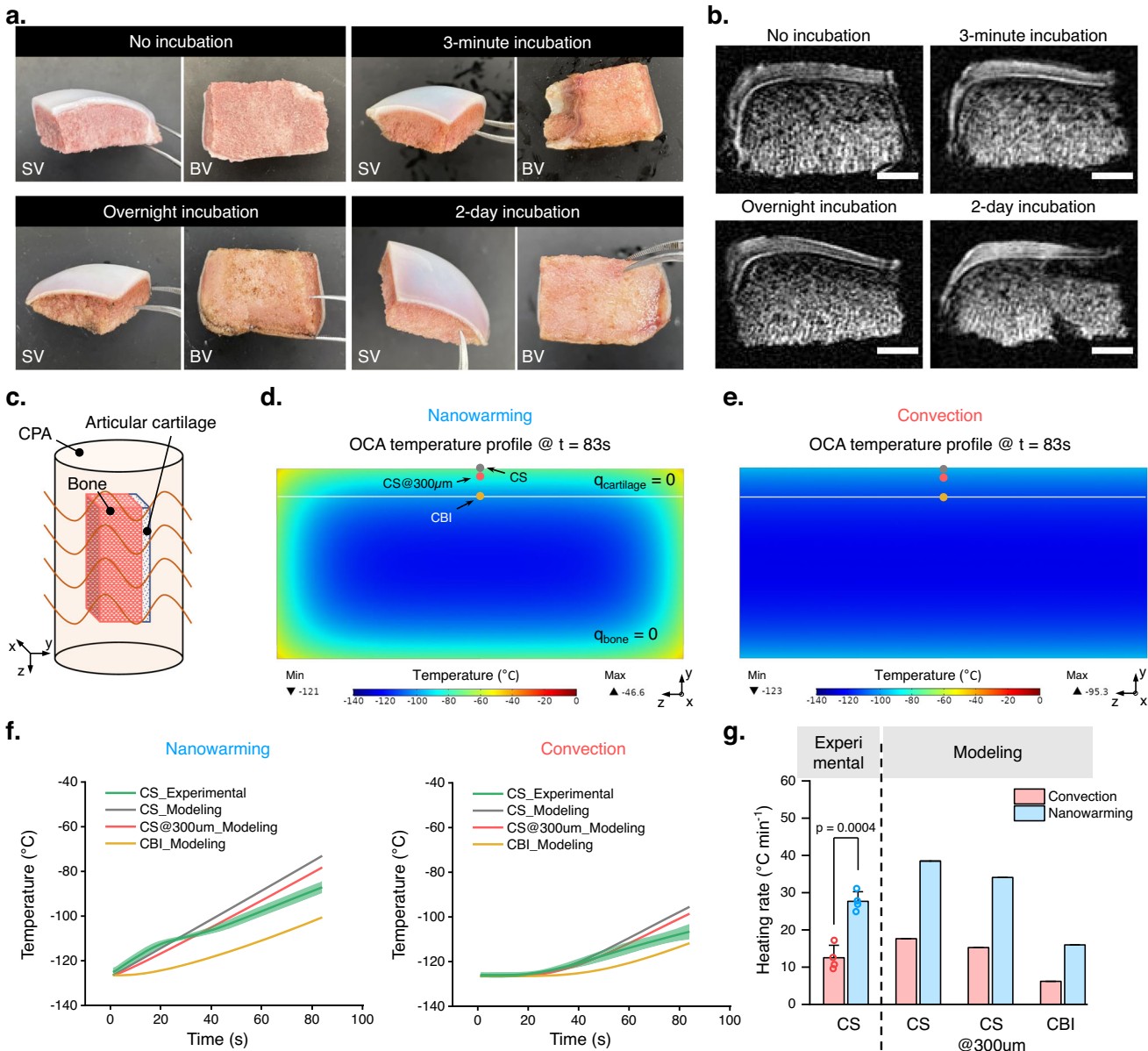

**Fig. 5 Computational thermal modeling reveals depth-dependent warming profile during large OCA nanowarming. a** Visual inspection of OCA incubated in VS83 solutions with 2 mg ml$^{-1}$ mIONPs for 0 min (no incubation), 3 min, overnight (~12 h), and 2 days (~48 h). SV side view, BV bone-side view. **b** MRI scans of OCA incubated in VS83 solutions with 2 mg ml$^{-1}$ mIONPs for 0 min (no incubation), 3 min, overnight (~12 h), and 2 days (~48 h) using gradient echo sequence. Scale bars represent 4 mm. **c** Geometry of the OCA sample and the tube for the nanowarming computational modeling. The tube was filled with the CPA solutions. The tube dimensions: 14 mm in radius and 63 mm in height (~39 ml); the OCA dimensions: 20 mm in width, 30 mm in length, and 14 mm in depth (~9 ml). **d** Temperature profile at the cross-sectional plane of the model system following 83 s of nanowarming. $q_{cartilage}$ and $q_{bone}$ are the heat generation rates in the cartilage region and bone region, respectively. The sites on the cartilage surface (CS), 300 μm underneath the cartilage surface (CS@300 μm), and cartilage-bone interface (CBI) are indicated with gray, red, and yellow dots. **e** Temperature profile at the cross-sectional plane of the modeling system following 83 s of convection warming. **f** Temperature profile on the CS, CS@300 μm, and CBI for both nanowarming and convection warming. The green line and light green shaded ribbon show the average temperature value collected from the experimental measurements at the cartilage surface and its standard deviation, respectively ($n = 4$ independent samples). The gray, red, and yellow lines are the modeling data. **g** Heating rate on the CS, CS@300 μm, and CBI derived from the temperature profile presented in (**e**). Experimental data is measured from $n = 4$ independent samples. $p$-value was determined with a two-sided $t$-test. All data depict mean ± standard deviation. Note: mIONPs in the nanowaming group are loaded through the immersion approach for all data presented here.

solution via nanowarming and the cartilage was warmed up through heat transfer from the surrounding solution to the cartilage surface and then the deep region of the cartilage. However, if the mIONPs can be delivered to the bone region of OCAs, there will be heat generated within the bone region. The heat can then be transferred to the cartilage through the heat pathway at the bone side of the cartilage, therefore resulting in faster cartilage

rewarming. To test the feasibility of this idea, direct mIONP injection to the bone side was performed (Fig. 6a). Visual inspection showed the successful delivery of mIONPs into the OCA bone region (Fig. 6b), and MRI results validated the presence of mIONPs in the injection region (Fig. 6c). We then performed computational modeling to evaluate the effect of the mIONP delivery on OCA nanowarming. We implemented the

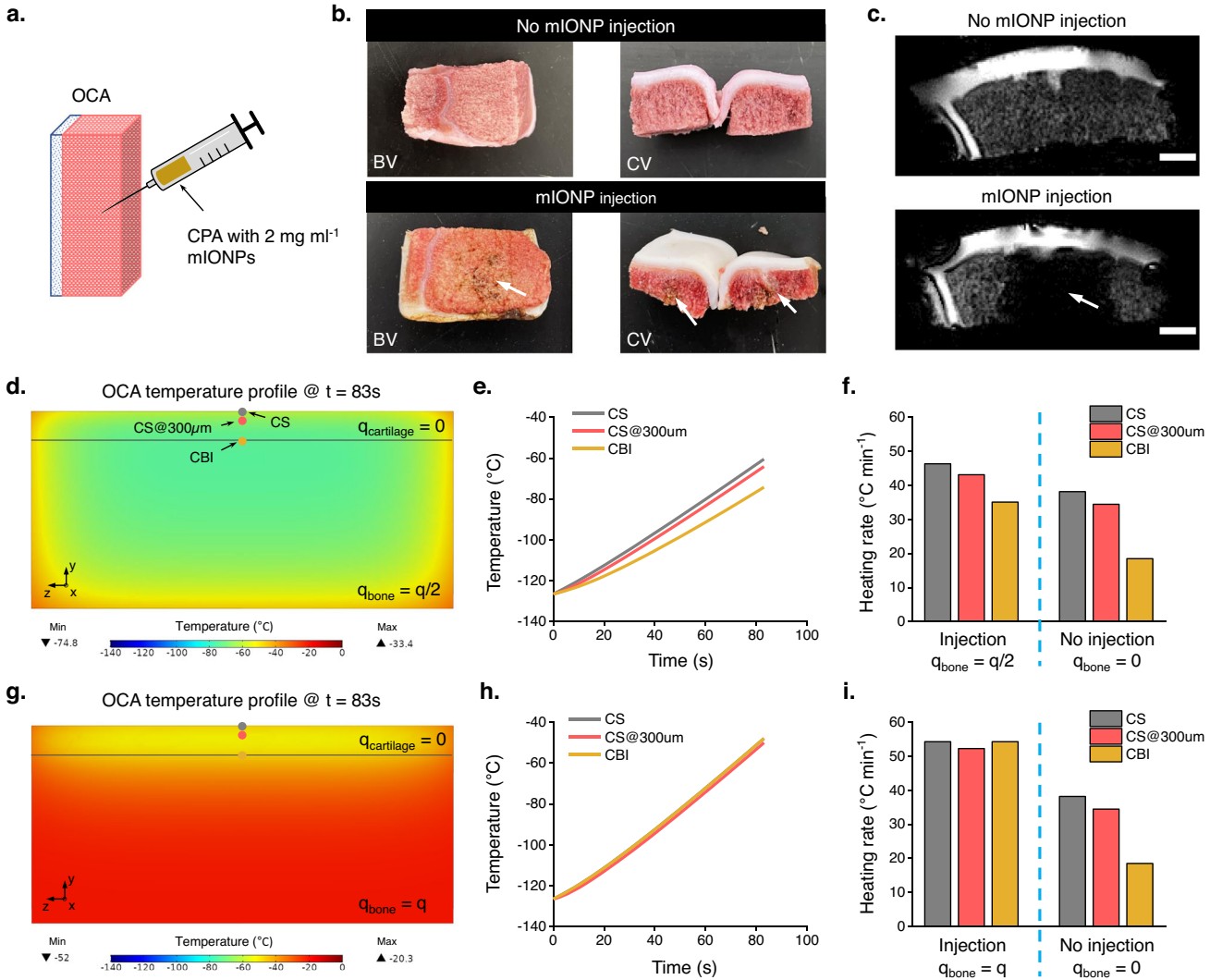

**Fig. 6 Delivery of mIONPs into the OCA bone region helps the preservation of large-sized cartilage. a** Schematic of mIONP delivery to the bone region of OCA through injection. **b** Visual inspection of OCA samples with and without mIONP injection. The white arrow highlights the injection region with a brownish color. BV bone-side view, CV cross-sectional view. **c** MRI scans of OCA with and without mIONP injection using gradient echo sequence. The white arrow highlights the injection region with diminished MRI signals due to the presence of mIONPs. Scale bars represent 4 mm. **d** Temperature profile at the cross-sectional plane of the model system following 83 s of nanowarming when the heat generation rate in the bone region is half of that in the CPA surrounding the OCA. $q_{cartilage}$ and $q_{bone}$ are the heat generation rates in the cartilage region and bone region, respectively. The sites on the cartilage surface (CS), 300 μm underneath the cartilage surface (CS@300 μm), and cartilage-bone interface (CBI) are indicated with gray, red, and yellow dots. **e** Temperature profile on the CS, CS@300 μm, and CBI shown in (**d**). **f** Heating rate on the CS, CS@300 μm, and CBI derived from the temperature profile presented in (**e**). Heating rate results from the nanowarming modeling without mIONP injection to the bone region were also presented for comparison. $q_{bone}$ and $q$ are the heat generation rate in the bone region and in the CPA solution, respectively. **g** Thermal profile at the cross-sectional plane of the model system following 83 s of nanowarming when the heat generation rate in the bone region is equal to that in the CPA surrounding the OCA. **h** Temperature profile at the site of the CS, CS@300 μm, and CBI shown in (**g**). **i** Heating rate at the site of CS, CS@300 μm, and CBI derived from the temperature profile presented in (**h**). Heating rate results from the nanowarming modeling without mIONP injection to the bone region were also presented for comparison. $q_{bone}$ and $q$ are the heat generation rate in the bone region and in the CPA solution, respectively. Note: mIONPs in the nanowarming group are loaded through the injection approach for all data presented here.

same computational model used earlier (Fig. 5c) and introduced a heat source term in the bone region. The value of the heat source term at the bone region was defined as either half (Fig. 6d–f) or equal (Fig. 6g–i) to that of the surrounding CPA solution in the two cases shown here, respectively. These two concentrations were selected to reveal that the warming rate at the deep region of the cartilage increases with increasing nanoparticle concentrations on the bone side, therefore guiding future protocol optimization. The results showed that heat transfer within the cartilage region is more efficient and that the heating rate increases with the rise of the heat generation level at the OCA

bone region (Fig. 6d–i). Therefore, enhancing the distribution of mIONPs in the OCA bone regions could be a feasible means to further improve the nanowarming and ice-free cryopreservation method for large OCA preservation.

## Discussion
This study initiates the cryopreservation of large OCAs. By incorporating vitrification and nanowarming, our method enables faster sample rewarming compared to convection warming, thereby reducing the potential for sample injury due to ice formation. Quantitative analysis of cartilage chondrocyte cell

metabolic activity and cell viability revealed that our method outperforms the traditional convection strategy in preserving large OCAs. Tissue level characterizations also showed that our method did not adversely affect cartilage material and mechanical properties. These results demonstrate the feasibility of this nanowarming and ice-free cryopreservation method for large-size OCA cryopreservation.

Long-term OCA cryopreservation is critical to improve donor OCA utilization and alleviate the OCA shortage. Current studies on cartilage preservation mainly focus on the vitrification process. Different vitrification protocols with various CPA solutions, loading times and steps, and loading temperatures have been proposed[24–26,45,46]. However, the warming procedure has not been well investigated. The use of a warm water bath is the current standard for sample rewarming. In the case of small OCA cryopreservation, the warming process is less critical to cartilage preservation success since convection warming is sufficient for fast rewarming. However, with larger samples, rewarming becomes more challenging due to heat transfer limitations. It has been reported that the heating rate slumps when the sample size (tissue plus surrounding solutions) exceeds 10 mm in diameter[28]. Without new protocols or strategies, it is impossible to meet the heating rate requirement for large, vitrified sample rewarming employing the traditional convection warming method. Our method incorporates nanowarming and vitrification, providing a feasible means to resolve the warming issues. Our data revealed the inability of the traditional convection method for large OCA preservation and showed that improving the heating rate is needed to achieve large cartilage cryopreservation. Our findings are aligned with a recent literature report[47]. In this prior literature study, CPA solutions surrounding the condyle were depleted, enabling faster heat transfer due to direct contact between the cartilage and water bath[47]. In contrast, in our nanowarming method, the surrounding CPAs remained and were rapidly warmed through nanowarming to achieve fast heat transfer and an elevated heating rate. Compared to the CPA depletion method in literature, nanowarming has a smaller chance of generating a significant thermal gradient across the samples as the sample surface is not immediately heated to a high temperature and is flexible for heating efficiency improvement by delivery of nanoparticles to the bone region as shown here (Fig. 6). Additionally, retaining the surrounding CPA solution might help isolate the OCA, therefore preventing OCAs from having direct contact with other surfaces and potentially avoiding damage during storage or transportation. A comprehensive cell and tissue level evaluation has shown the benefits of our method over the convection method. There is a great clinical need for large OCA cryopreservation to assist in surgical planning by providing additional time for donor–recipient matching and screening. Moreover, compared to small OCAs, large, preserved OCAs provide the flexibility to harvest grafts of any size and shape to fit the patient's defect site at the time of operation.

Nanowarming is a promising technique for large tissue and organ preservation. In this study, we demonstrated the application of nanowarming for the preservation of porcine OCAs with a tissue volume of approximately 9 ml (a total volume of ~39 ml including the surrounding 30 ml CPA), which is much larger than previously utilized tissues and organs reported in the literature[30–32,38]. Nanowarming has been applied to preserve porcine arteries, porcine heart valves, rat hearts, and rat kidneys; rat hearts and kidneys have relatively small volumes, ≤1 ml, while porcine arteries and porcine heart valves are both thin. Undeniably, these literature reports have successfully introduced the concept of nanowarming and proved the capacity of nanowarming for the preservation of biological samples, especially complex organs such as the heart and kidney. However, the scale-up feature of nanowarming for large-sized tissue and organ preservation has not been yet demonstrated. Here, we implemented nanowarming with a large piece of biological tissue and systematically evaluated the outcome of nanowarming for large tissue preservation, moving the nanowarming technique toward clinical practice. In addition, articular cartilage is a dense tissue and has no blood vessel network. In contrast to vascularized systems like the heart and kidney, mIONPs cannot be uniformly distributed within the tissue through immersion or perfusion; visual inspection and MRI scans showed that no mIONPs were present inside the tissue (Fig. 5a, b and Supplementary Fig. 6). Although uniform warming could not be achieved due to the absence of mIONPs inside the cartilage, the solution surrounding the cartilage can be quickly warmed up and transfer the heat to the cartilage. The resulting heating rate inside the tissue reached 16.0–38.5 °C min$^{-1}$, which is much higher than the 6.2–17.6 °C min$^{-1}$, achieved through convection warming. The nanowarming heating rate must be high enough to avoid cartilage tissue damage during the rewarming process. This notion was supported by both in vitro experiments and modeling results. Our results indicate that nanowarming helps better preserve the cartilage of large OCAs, particularly at the surface layer, and that nanowarming is a feasible approach for the preservation of dense and avascular tissues.

AlarmarBlue test results demonstrated that increasing CPA concentration dramatically enhanced chondrocyte metabolic activities for both warming methods. Higher CPA concentration usually reduces critical cooling/warming rates, therefore reducing the chance of ice formation during both cooling and warming processes; this might explain the observed benefit of high concentration CPA (VS83) used in this study for OCA preservation over the lower concentration CPA (VS55 and VS70) (Fig. 2a and Supplementary Fig. 3). Cell metabolic activity results showed better cell function recovery in the nanowarming group than the convection group (normalized cell metabolic activity at Day 3 tissue culture: nanowarming vs convection in VS83 group, 102 ± 14% vs. 63 ± 12%, respectively). Our live/dead staining results also demonstrated that nanowarming rescued more live cells than convection warming (cell viability: nanowarming vs. convection, 46.1 ± 15.9% vs. 26.4 ± 9.8%, respectively). However, the cell viability of nanowarmed cartilage was different than that of fresh cartilage. The live/dead staining results showed that nanowarmed cartilage had very high green fluorescence at the superficial layer of the cartilage but dim green signals at the middle zone and deep zone of the cartilage, while fresh cartilages exhibited homogeneous green signals at all depths. Cells in the middle and deep zones of nanowarmed cartilage were not as active as the cells in fresh cartilage (i.e., with less enzyme to interact with calcein AM to produce a green fluorescence signal). The different live/dead staining results at different cartilage depths might be correlated with the depth-dependent heating rate. The cartilage layer with a higher heating rate had more live and functional chondrocytes. The cell viability analysis was based on measurements immediately after rewarming. Future studies should investigate cell viability after cartilage tissue culture to better compare differences between the nanowarmed and fresh cartilage.

Articular cartilage is a complex mixture of water and extracellular matrix components. As cartilage conductivity and FCD are important cartilage material properties regulating electromechanical behaviors, quantitative analysis is needed to evaluate cryopreservation efficacy. Fresh cartilage conductivity determined in this study agrees well with that measured by other groups[18,48]. A previous study has shown that cartilage conductivity decreases following 28-day hypothermic storage[18]. Our results showed no significant changes in cartilage conductivity

after cryopreservation with convection and nanowarming methods, indicating that our method has a very limited impact on cartilage electrical behaviors. Moreover, we observed comparable FCD in nanowarmed cartilage to fresh cartilage but decreased FCD in the convection-warmed cartilage. This result suggests that the convection-warmed cartilage might lose its glycosaminoglycan content and lean towards matrix degradation[49] and therefore not be suitable for implantation.

Few studies have investigated the impact of cryopreservation on cartilage mechanical properties. By performing both local and full-thickness mechanical tests, our results suggested depth-dependent changes in cartilage mechanical properties following cryopreservation. At the cartilage surface layer, nanowarmed cartilage has comparable mechanical properties to fresh cartilage. The sample size was ~30 mm × 20 mm × 14 mm (length × width × thickness) in our study, which was much larger than those used in prior literature reports. Kiefer et al. used small bovine articular cartilage (6 mm diameter, 3 mm thickness) to investigate the effect of cryopreservation on mechanical behavior and found no significant changes after cryopreservation[50]. Li et al. also demonstrated that vitrified porcine cartilage (9 mm in diameter) had comparable mechanical properties to fresh cartilage[51]. Our results with large-sized samples agreed with these prior literature findings with small samples[50,51], indicating the scalable capacity of our method for cartilage cryopreservation. However, nanowarmed cartilage showed compromised full-thickness mechanical properties, suggesting the need for further optimization of the nanowarming protocol. As shown in the modeling work, delivery of mIONPs to the bone region of OCAs could be one possible solution. Future studies can experimentally evaluate the performance of this strategy and investigate related methods to wash out the mIONPs.

Our nanowarming modeling results at the cartilage surface showed some differences with the experimental measurements at later times (Fig. 5f). The discrepancies between the experimental and modeling results could be caused by using temperature-independent CPA thermal properties in our model. It has been shown that the CPA thermal parameters, such as the specific heat, thermal conductivity, and density, are temperature dependent[28,31,32,52–54]. The nanowarming heat generation rate has also been reported to be temperature dependent[31,32,52]. All these parameters could affect the model predictions and therefore need to be determined experimentally. In our study, a simplified model was built and averaged CPA thermal properties as used in a previous report[30,55] were applied. Utilizing this simplified model, we can reveal possible causes of our experimental depth-dependent preservation observations and hinted potential means to improve the OCA preservation outcome further. Future studies are certainly needed to build a more comprehensive model representing the actual nanowarming with the OCA sample. Given the limited data on the VS83 thermal properties, it will also be critical to establishing the database of different CPA thermal properties in the cryopreservation field.

In this study, we mainly focused on the warming process. However, multiple factors affect cryopreservation outcomes, including various settings for the cooling and warming steps. Future studies can incorporate the latest cryopreservation progress to enhance CPA loading and reduce CPA toxicity to further improve cryopreservation outcomes. For instance, adding additives to the CPA solutions may further improve results since additives, chondroitin sulfate, and ascorbic acid, have been shown to mitigate the toxicity of high cryoprotectant concentrations[26]. Additionally, the nanowarming protocol can also be further tuned with the assistance of computational modeling and advanced fabrication techniques to make more stable, biocompatible, and efficient heat-generating nanoparticles[31,39,45].

With further optimization of the nanowarming protocol, this biobanking method for cartilage preservation will greatly improve the utilization of the limited supply of allografts, increase long-term allograft survival in vivo, and expand the clinical applications of osteochondral allograft transplantation.

## Methods

**Sample preparation.** Fresh femoral trochlea of sexually mature (mixed sex) domestic Yorkshire cross pigs (4–6 months old) were obtained from a local slaughterhouse and used for the preservation procedure ~4 h post-mortem. The use of animal tissues is approved by Clemson University and the Medical University of South Carolina. No animals were specifically sacrificed for this study. Osteochondral allografts (OCAs) of the femoral trochlea (length × width × thickness, ~30 mm × 20 mm × 14 mm), representing ~25% of the trochlea, were harvested (Fig. 1a) and placed in sterile cups containing cold Dulbecco's modified Eagle's medium (DMEM; Corning, Arizona) supplemented with 100 IU per ml penicillin and 100 μg ml$^{-1}$ streptomycin (Millipore Sigma, MO). They were then randomly separated into three groups: fresh, vitrification with convection warming, and vitrification with nanowarming.

**Vitrification procedure.** Three different vitrification solutions were tested in this study: commonly used VS55 (3.10 M DMSO, 3.10 M formamide, and 2.21 M 1,2-propanediol), our recently developed VS83 (4.65 M DMSO, 4.65 M formamide, and 3.31 M 1,2-propanediol)[25], and VS70 (3.97 M DMSO, 3.97 M formamide, and 2.83 M 1,2-propanediol), a CPA solution with a concentration between VS55 and VS83. OCAs were gradually infiltrated with VS83, VS70, or VS55 in EuroCollins solution at 4 °C. The precooled diluted vitrification solutions were added in six sequential 20-min steps of increasing concentration (0%, 18.5%, 25%, 50%, 75%, and 100%) on ice (Fig. 1b). Each specimen was then transferred to a 50-ml centrifuge tube containing 30 ml of precooled 100% vitrification solutions (4 °C) with or without 2 mg Fe per ml (Ferrotec, EMG-308) for nanowarming and convection groups, respectively. Then, all samples were immediately placed in a precooled 2-methybutane bath at −135 °C and vitrified[25]. Specimens were stored at −135 °C in a mechanical storage freezer for at least one week before rewarming. The mIONP loading protocol in the nanowarming group described here was defined as an immersion loading method since the CPA-loaded OCAs were transferred into the CPA with mIONPs and then immediately vitrified.

**Convection warming.** Vitrified OCAs in the convection warming group was warmed by placing the plastic container with specimens in a water bath at 37 °C. After rewarming, the vitrification solution was removed in seven sequential 20-min steps at 0 °C into the DMEM culture medium.

**Nanowarming.** Vitrified OCAs in the nanowarming group was warmed in ~ 80 s using a 5.0 kW terminal, 230 kHz, EASYHEAT 5060LI solid-state induction power supply with a remote work head and custom multi-turn helical coil with 6–7 turns to create a 35 kA m$^{-1}$ magnetic field in the center. After rewarming, the mIONPs and vitrification solution were washed off in a similar stepwise manner as described for the specimens in the convection warming group.

**Temperature measurement in the CPA solutions and OCA in CPA solutions.** The temperature–time curves in both cooling and warming processes were measured by a fiber optic signal conditioner (FOTEMP1-H, Micronor INC, Camarillo, CA) coupled with a fiber-optic temperature sensor (TS3, Micronor INC, Camarillo, CA). For the temperature measurement in the CPA solutions (both VS55 and VS83), the fiber optic temperature sensor was well secured along the axis of the 50 ml Falcon tube (Supplementary Fig. 2a). The sensor tip was positioned in the middle of the filled CPA solutions to determine the temperature profile at the center of the CPA solution (Supplementary Fig. 2a). For the temperature measurement at the cartilage surface of OCAs immersed in the VS83 with/without mIONPs (for convection warming and nanowarming, respectively), the sensor tip was well positioned at the center of the cartilage surface. The cooling temperature curve was measured while the samples (pure CPA solutions or OCA in VS83) were vitrified following the cooling procedure. The cooling rate was calculated as the first derivate of the temperature–time curve at each time point. For the warming temperature curve, the temperature rise was recorded during the process of convection warming and nanowarming. The heating rate was determined as the mean heating rate (i.e., the amount of temperature rise divided by the time) of the first 10–60 s of temperature data.

**Cell metabolic activity measurement by alamarBlue assay.** Following the rewarming process, OCAs were collected for the cell metabolic activity test. The bone side of the OCA was first removed. The remaining articular cartilage specimens were then cut into small pieces and incubated in 2 ml of DMEM for one hour to equilibrate, followed by the addition of 10% alamarBlue (Thermo Fisher Scientific, Waltham, MA) under standard cell culture conditions for 3 h. The amount of fluorescence of each cartilage piece was measured in 12–24 duplicates by the

multimode microplate reader at an excitation wavelength of 544 nm and an emission wavelength of 590 nm. As alamarBlue is not cytotoxic, cartilage specimens from the convection and nanowarming groups were cultured in the medium for 3 days, and the same fluorescence measurements were performed daily to allow the characterization of cell activity changes over time. After the final fluorescence was read, cartilage specimens were dried and weighed. The average fluorescence read of blank samples (only alamarBlue solution without cartilage specimens) was subtracted from the fluorescence read of each cartilage piece and then divided by the cartilage dry weight to yield relative fluorescent units per mg of dry tissue. Results from the convection and nanowarming groups were normalized to the results measured from fresh cartilage at day 0 to obtain percentages.

**Fluorescence live–dead cell staining**. Cell viability was examined using fluorescence live/dead staining. Full-thickness cartilage strips were manually cut out from the cartilage specimens with a razor blade and immersed in DMEM culture medium with 2 μM calcein AM (Thermo Fisher Scientific, Waltham, MA) and 4 μM ethidium homodimer-1 (Thermo Fisher Scientific, Waltham, MA) at room temperature for 1 h before imaging. Calcein AM is cell membrane-permeable and interacts with the cytoplasmic esterases in live cells to generate a green fluorescence signal. Ethidium homodimer-1, instead, is membrane-impermeable, and it enters dead cells with damaged cell membranes and binds with nuclear acid to produce a red fluorescence signal. Then, the stained cartilage strips were placed on a glass-bottom petri dish with its cross-sectional side laying over the glass. An inverted Olympus FV1200MPE multiphoton laser scanning microscope (Olympus, Center Valley, PA) with a ×10 lens was used for imaging. The excitation wavelength was 800 nm[56]. The green fluorescence signal was collected within the 495–540 nm range while red fluorescence signal was collected in the 575–630 nm range. Cross-section images were taken at the depth of 100 μm below the tissue surface. After fluorescence imaging, cell viability in the cartilage surface layer (within ~400 μm from the cartilage surface) was quantified using ImageJ (V1.52a, National Institutes of Health, Bethesda, MD). The intensity of fluorescence signals from the green and red channels is used to differentiate between live and dead cells. Cells with higher green fluorescence intensity were counted as live cells while cells with higher red fluorescence intensity were counted as dead cells. Cell viability was defined as the ratio between the number of live cells and the total number of cells (live cell number plus dead cell number).

**Histology analysis**. Osteochondral plugs collected from the three experimental groups (fresh, convection, and nanowarming) were fixed in 10% formalin solution, dehydrated, and then paraffin embedded. 5 μm sections were cut using a microtome and processed for the Hematoxylin & Eosin (H&E) staining and Safranin O staining. Bright-field images were acquired with ×4 and ×10 objectives on an Olympus BX40 microscope (Olympus, Center Valley, PA).

**Electroconductivity measurement**. A customized conductivity apparatus (Fig. 3a) was used to examine tissue electrical conductivity and determine cartilage fixed charge density (FCD)[43,44,57]. The conductivity chamber consists of current sourcing and voltage-measuring electrodes placed around a Plexiglas chamber containing each specimen. Employing a Keithley Source Meter (Model 2400, Keithley Instruments, Inc., Cleveland, OH), the resistance ($R$) across the specimen was determined at a very low current density of 0.015 mA cm$^{-2}$. The corresponding electrical conductivity ($\chi$) was calculated using the following equation:

$$\chi = \frac{h}{RA} \tag{1}$$

where $h$ and $A$ are the height and cross-section area of the specimen, respectively. The specimen thickness, $h$, was measured using a current sensing micrometer immediately before conductivity measurement. FCD was determined by a two-point electrical conductivity method. Cartilage specimens were punched into 5-mm cylindrical plugs with a trephine and trimmed on both sides to ensure the surface was flat using a microtome. Specimens were then immersed in isotonic KCl solution (0.15 M) for 12 h at 4 °C while axially confined between two hydrophilic polyethylene porous platens (50–90 μm; Small Parts, Inc., Miami Lakes, FL) in a 5 mm diameter chamber. After incubation, cartilage electrical conductivity was measured ($\chi_{Iso}$). Immediately after measurement in an isotonic solution, specimens were incubated in hypotonic KCl solution (0.03 M) for 12 h and cartilage conductivity ($\chi_{Hypo}$) was measured again. Specimen FCD ($c^F$) was then determined by the two conductivities measured at the isotonic and hypotonic conditions[43,44]:

$$c^F = 2\sqrt{\frac{\chi_{Iso}^2(c_{Iso}^{*\,2} - c_{Hypo}^{*\,2})}{\chi_{Iso}^2 - \chi_{Hypo}^2} - c_{Iso}^{*\,2}} \tag{2}$$

where $c_{Iso}^*$ and $c_{Hypo}^*$ are the ion concentrations of the isotonic and hypotonic KCl solutions, respectively. All electrical conductivity measurements were performed at room temperature (22 °C).

**Porosity measurement**. Cartilage specimen porosity was measured using the buoyancy method. Immediately before specimen incubation in an isotonic solution, specimen weight was recorded in air ($W_{wet}$) and in phosphate-buffered saline

($W_{PBS}$) using a density determination kit and analytical balance (Sartorius YDK01, Germany). Immediately after conductivity measurement in hypotonic condition, specimens were lyophilized and the dry weight ($W_{dry}$) was recorded. The porosity ($\phi^W$) was determined by:

$$\phi^W = \frac{W_{wet} - W_{dry}}{W_{wet} - W_{PBS}} \frac{\rho_{PBS}}{\rho_w} \tag{3}$$

where $\rho_{PBS}$ and $\rho_w$ are the density of the PBS solution and water, respectively.

**Mechanical testing**. Microindentation tests were performed using a bioindenter system (UNHT³ Bio, Anton Paar, Switzerland) (Fig. 4a). The resolution for the force and displacement were down to 0.001 μN and 0.006 nm, respectively. One osteochondral plug in 5 mm diameter was punched out at the center of each individual sample. Four equally spaced thickness measurements were performed using a calibrated stereoscope, and the average value was defined as the cartilage thickness. Following thickness measurement, osteochondral plugs were perpendicularly mounted on the sample holder with the bone end carefully glued to a Petri dish using a small amount of cyanoacrylate. PBS solution was then added to fully immerse the sample. The sample holder was well positioned to ensure the indenter was perpendicular to the cartilage surface. A spherical ruby ball indenter (1 mm diameter) was used for testing. At the start of the creep test, a 0.05 mN tare load was applied and held for 600 s, followed by a 5 mN step load for 800 s to generate the creep data. The indentation force and displacement data were acquired at 20 Hz. Two to three indentation tests spaced ~500 μm apart were performed at the central region of each sample. The creep data were analyzed with the Hertzian biophysical theory to obtain the equilibrium contact modulus and the hydraulic permeability at the cartilage surface[58].

Confined compression mechanical tests with the cartilage specimens were performed with a dynamic mechanical analyzer (DMA Q800, TA Instruments, Delaware, USA) (Fig. 4d). Precision for the force and displacement measurements was 0.1 mN and 0.1 μm, respectively. Cartilage specimens for the mechanical tests were first punched into a 5 mm cylindrical plug with a trephine and trimmed flat using a microtome. Then 5 mm cartilage specimens were axially compressed between a testing probe (5 mm diameter) on top and a rigid porous permeable platen (~20 μm) on the bottom. Specimens were first compressed with a tare load (0.05 N) to ensure contact between the specimen and testing probe and measure the initial specimen height. PBS was then carefully added to the chamber and 10% compressive strain (relative to initial height) was applied. The specimens were allowed to reach equilibrium (~1 h) and equilibrium stress was recorded. Using the biphasic theory[59], the equilibrium compressive aggregate modulus and hydraulic permeability coefficient were determined by fitting the stress relaxation curve.

**MRI scans**. Fresh OCAs were first immersed in VS83 solutions without mIONPs for 2 h and then incubated in VS83 solutions with 2 mg ml$^{-1}$ mIONPs for different amounts of time (3 min, overnight, and 2 days) before MRI scans. OCAs immersed in VS83 solutions without mIONPs was used as the control group. Prior to MRI scanning, the excess CPA solutions surrounding the OCAs were removed and the OCAs were wrapped with plastic wrap. A 7 T MRI scanner (BioSpec 70/30 USR, Bruker Corp., Ettlingen, Germany) was then used to scan the OCA samples with both gradient echo sequence and spin echo sequence. 1% agarose gels with different mIONP concentrations were prepared as standard phantoms to show the MRI contrast with the presence of mIONPs and were scanned with the identical sequences used for OCA imaging. Scanning settings for the gradient echo sequence were: repetition time/echo time (TR/TE) = 200/2.5 ms, flip angle = 30°, the field of view = 40 mm × 30 mm, matrix size = 256 × 192, in-plane resolution = 0.156 mm × 0.156 mm; slice thickness = 1.0 mm with 0.3 mm gap, total scanning time = 1.5 min. Scanning protocol for the spin echo sequence was: TR/TE = 2000/5 ms, flip angle = 90°, the field of view = 40 mm × 30 mm, matrix size = 128 × 128, in-plane resolution = 0.313 mm × 0.234 mm; slice thickness = 1.0 mm with 0.3 mm gap, total scanning time = 4.5 min.

**Computational modeling**. The nanowarming process was simulated using finite element analysis software COMSOL (V5.4, COMSOL Inc, Burlington, MA). Two computational models were built to simulate nanowarming with the VS83 solutions only and nanowarming with OCA samples in the VS83 solutions. The model for the VS83 solution nanowarming was simplified as a cylindrical tube (14 mm × 49 mm, Radius × Height) filled with CPA solution with mIONPs, corresponding to 30 ml of the CPA solution used in the actual experiment (Supplementary Fig. 5g). The computational model for the OCA samples consisted of a cylindrical tube (14 mm × 63 mm, Radius × Height) and the osteochondral block (30 mm × 20 mm × 14 mm, Length × Width × Thickness) (Fig. 5c). Cartilage thickness was defined as 2 mm[45]. The thermal profile was governed by the heat transfer equation:

$$\rho C_P \frac{\partial T}{\partial t} = \nabla \cdot [k \nabla T] + q \tag{4}$$

where $\rho$ is the density, $C_P$ is the specific heat, $k$ is the thermal conductivity, $t$ is the time, $T$ is the temperature, and $q$ is the heat generation rate due to the nanowarming. Following prior literature[31,32], CPA solution properties were used instead

of the CPA-permeated osteochondral graft in this model. There is very limited VS83 thermal data in the literature. As our results indicated that VS83 has similar thermal properties to VS55 (Supplementary Fig. 5a–f), the thermal parameters of VS55 solution ($\rho = 1100\,\text{kg m}^{-3}$, $C_\text{p} = 2.1 \times 10^3\,\text{J kg}^{-1}\,\text{K}^{-1}$, $k = 0.3\,\text{W m}^{-1}\,\text{K}^{-1}$)[30,55] were applied in this study. The heat generation term, $q$, in the OCA was defined as zero due to the absence of the nanoparticles (Fig. 5a, b and Supplementary Fig. 6). The heat generation rate in the surrounding CPA solution was determined and validated through nanowarming experiments with CPA solutions with mIONPs. The boundary condition for the tube was assumed to be insulated[28]. For the simulation of two cases of nanowarming with nanoparticles delivered into the bone region of the OCA, the heat generation rate at the bone region was set as either half (Fig. 6d–f) or equal (Fig. 6g–i) to that of the surrounding CPA solution. The rest of the parameters were kept the same as in the prior simulation. Convection warming was also modeled. Although the heat transfer coefficient between the tube and warm water bath was not determined in this study, we set a constant tube surface temperature, 37 °C (i.e., infinite heat transfer coefficient), to model the highest heating rate of convection warming. All models use extra fine mesh settings in COMSOL.

**Statistics and reproducibility**. A two-sided $t$-test was performed to evaluate the heating rate difference between convection warming and nanowarming. A one-way ANOVA with Bonferroni post-hoc test was used to examine differences in fluorescence imaging-based cell viability, electrical conductivity, FCD, porosity, and mechanical properties of cartilage specimens among the fresh, convection, and nanowarming groups. Since multiple measurements were performed at different locations of the same sample in the microindentation test, measurement location was defined as a covariate for the statistical analysis. All analyses were performed in SPSS (IBM SPSS Statistics, Version 24.0, IBM Corp., Armonk, NY). Significant differences were reported at $p < 0.05$ with descriptive statistics reported as mean ± standard deviation.

**Reporting summary**. Further information on research design is available in the Nature Portfolio Reporting Summary linked to this article.

## Data availability

The raw data that support the findings of this study are available from the corresponding author upon reasonable request. Source data for the figures are provided in the Supplementary Data in this paper.

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

## Acknowledgements

This work was supported by National Institutes of Health (NIH) grants R44AR073136 to K.G.M.B., P20GM121342, and R01DE021134 to H.Y. This work was also supported by NIH T32 post-doctoral fellowship DE017551 to R.G.H. and C.H., and NIH grant K99DE031345 to C.H. This work was also supported in part by the Cell & Molecular Imaging Shared Resource, Hollings Cancer Center, Medical University of South Carolina (P30CA138313), the SC COBRE in Oxidants, Redox Balance, and Stress Signaling (P20GM103542), and the Shared Instrumentation Grant S10OD018113.

## Author contributions

The manuscript was written through contributions by all authors. All authors have given approval for the final version of the manuscript. H.Y., K.G.M.B., and P.C. conceived the idea and designed the experiments. P.C., S.W., Z.C., P.R., R.G.H., and Y.W. performed the cell-level and tissue-level experiments. Z.C., E.D.G., L.H.C., P.C., and S.W. performed the vitrification and nanowarming processes. K.L.H., P.C., and S.W. conducted the histology analysis. P.C., X.N., and J.H.J. performed the MRI scans. P.C. and H.Y. analyzed the data. P.C. and H.Y. drafted the manuscript. P.C., S.W., Z.C., C.H., K.G.M.B., and H.Y. reviewed and edited the manuscript.

## Competing interests

K.G.M.B., Z.C., E.D.G., and L.H.C. are employees of Tissue Testing Technology LLC. The remaining authors declare no competing interests.
