## [Peer Review File · Communications Biology]

Reviewers' comments:

Reviewer #1 (Remarks to the Author):

This work looks at techniques to warm samples cryopreserved through a vitrification technique. Namely, if the addition of nanoparticles which can be excited to induce heating through use of a magnetic field improved post-warming performance. The work has merit in a range of clinical applications, with this work focusing on osteochondral allograft tissue. The group have measured a wide range of parameters to assess success of the method and so the reader can be confident in the robustness of results.

I recommend publication of this work with some corrections.

1. The technique for loading the nanoparticles in this work should be addressed more clearly – the injection method is mentioned at the end but is this what was used in the biological and histological data throughout? Are these particles chemically and biologically inert, and would they need to be washed out before the tissue was transplanted?

2. Did the authors measure the actual warming rate achieved in the tissue or rely on the modelling? The information in this regard given in Supplementary Note 2 is quite relevant and I think it would benefit from being included in the main text.

Minor corrections:

1. Most uses of "rewarmed" throughout the work should be changed to "warmed"
2. What is the advantage of this method over the microwave warming stated in line 94, stating this explicitly would help the impactfulness of the work.
3. Do the authors have a minimum warming rate in mind for those tissues cryopreserved in the high concentration CPA? Would future work to increase warming rate help or are the rates achieved here sufficient (with damage caused by other parts of the cryopreservation process)?
4. Line 382, remove word "in"
5. Line 430. I think "stripes" should read "strips"

Reviewer #2 (Remarks to the Author):

General comment:

The manuscript is very well written and clearly outlines the problem at hand. It explains the need for OCA preservation and its benefits to medicine. The authors have done a great job at outlining the methods for all the biological survival assays and mechanical strength testing experiments.

Specific comments:

1. The manuscript does not discuss or show the successful vitrification of the OCA in the current study. It would be beneficial to show an image and cooling rate profile of a vitrified OCA.
2. In Fig 2b, it seems like the nanowarmed OCA has a higher green fluorescence signal in the superficial zone compared to the fresh controls. Is there a reason for this to happen? Although the authors mention that nanoparticles do not penetrate in the OCA, it would be a good exercise to perform the assay on a control group wherein all the nanoparticle and CPA solution loading steps and unloading steps are performed without vitrification and rewarming to ensure there is no background signal from the nanoparticles.
3. The authors use a methyl butane bath at -135°C than the more commonly used liquid nitrogen bath at -196°C . What are the benefits of the current thermal protocol?
4. The concentration of mIONPs used in the current study is 2 mg/ml. Where studies performed to optimize the concentration?
5. Since the manuscript mentions the nanoparticles did not penetrate in the graft, did the authors look into other volumetric heating methods instead? How would nanowarming be a more optimal approach in this case compared to other volumetric heating methods?

6. The authors suggest delivery of mIONPs to the bone side of OCA for rapid rewarming. One challenge to this method is the mIONPs washout after rewarming. Have the authors developed a strategy for successful washout of mIONPs?
7. The concentration of mIONPs was assumed to be half or equal to that of the surrounding CPA solution in the bone side OCA for the nanoparticle delivery method. It would benefit the reader to know the reasoning behind assuming these concentrations.
8. The authors mention another study which was performed on a smaller cartilage by depleting the surrounding CPA solution. It would be interesting to see the comparison between the two methods for the same size of OCA, especially since the mIONPs do not penetrate the OCA. Would nanowarming also be superior in this case?
9. Fig 5b/6c: The contrast/LUT of the MRI scans can be calibrated and optimized for viewing of nanoparticles. It is difficult to observe on a grayscale. It would also benefit the reads to see one image with CPA loaded nanoparticles to be able to see the difference with and without nanoparticles.
10. Fig 5f-e: Due to the absence of mIONPs in the OCA, it would be interesting to see the temperature profile at the center of the bone in the computational model. Due to thermal inertia do the faster rewarming rates closer to the outer surfaces actually translate to the center of the bone? How much is the rewarming rate difference at the center using the two rewarming methods?
11. Have the modeling results been validated against the experimental results specifically with the OCA present? One way to do this would be to place thermocouples in the experimental set up and computational model at the same location and compute the difference.
12. Line 537: There 'is' very limited VS83 thermal data available in the literature.
13. Line 254-257: This statement is not entirely true due to the non-uniform temperature profile observed in the OCA. Thermal stresses can still be formed due to non-uniform heating. The key to reduce thermal stresses is uniformity in rewarming rates, which might still be a challenge due to the absence of nanoparticles in the OCA.

Re: COMMSBIO-22-2077-T

Nanowarming and ice-free cryopreservation of large sized, intact articular cartilage

Response to Reviewer #1:

This work looks at techniques to warm samples cryopreserved through a vitrification technique. Namely, if the addition of nanoparticles which can be excited to induce heating through use of a magnetic field improved post-warming performance. The work has merit in a range of clinical applications, with this work focusing on osteochondral allograft tissue. The group have measured a wide range of parameters to assess success of the method and so the reader can be confident in the robustness of results.

I recommend publication of this work with some corrections.

Response: We thank the reviewer for highlighting the significance of our work. We have responded to each of the reviewer's comments below.

1. The technique for loading the nanoparticles in this work should be addressed more clearly – the injection method is mentioned at the end but is this what was used in the biological and histological data throughout? Are these particles chemically and biologically inert, and would they need to be washed out before the tissue was transplanted?

Response: We thank the reviewer for raising this question. We would like to clarify that the immersion loading protocol (i.e., OCA was briefly immersed in the VS83 with nanoparticles after the VS83 loading steps and then immediately went through the cooling procedure) was used for all biological and histological data presented in the manuscript. The injection method is only applied to the data shown in Fig. 6 of the original manuscript (the same Fig. 6 was presented in the revised manuscript). The injection method was discussed as a potential future direction to further improve our technique. Our MRI data presented in Fig. 6 and Supplementary Fig. 6 of the revised manuscript showed that the nanoparticles could be delivered to the bone side via injection, and the modeling results in Fig. 6 further showed the benefits of the injection method to increase the heating rate at the deep region of the cartilage. We have added some text to clarify the loading protocol used for each dataset in the figure captions of the revised manuscript. We have also clearly defined the immersion loading protocol in the Method section (Line 440-442).

Iron oxide nanoparticles are regarded as inert nanomaterials and have been widely used in medical imaging and hyperthermia therapy¹. Literature studies have also demonstrated that the nanoparticles used in our study have no adverse effect on cell viability² and would degrade in the body over time³. However, to ensure safety, we would still suggest washing out the nanoparticles before transplantation. For the immersion loading method, the nanoparticles cannot penetrate the tissue and only reside at the OCA surface. They can be washed away during the CPA unloading process. For the injection method, we could wash out some of the nanoparticles

during the CPA unloading steps. Also, bone lavage, a procedure recommended to remove the bone marrow in the clinical OCA implantation surgery⁴, could help wash the nanoparticles out as well. Our future studies will investigate the optimized nanoparticle wash-out protocol for the injection method.

References:

1. Vallabani, N. V. S. & Singh, S. Recent advances and future prospects of iron oxide nanoparticles in biomedicine and diagnostics. *3 Biotech* **8**, 279 (2018).
<https://doi.org/10.1007/s13205-018-1286-z>
2. Hurley, K. R. *et al.* Predictable Heating and Positive MRI Contrast from a Mesoporous Silica-Coated Iron Oxide Nanoparticle. *Molecular pharmaceutics* **13**, 2172-2183 (2016).
<https://doi.org/10.1021/acs.molpharmaceut.5b00866>
3. Zhang, J. *et al.* Quantification and biodistribution of iron oxide nanoparticles in the primary clearance organs of mice using T(1) contrast for heating. *Magn Reson Med* **78**, 702-712 (2017).
<https://doi.org/10.1002/mrm.26394>
4. Sun, Y. *et al.* Pulsed lavage cleansing of osteochondral grafts depends on lavage duration, flow intensity, and graft storage condition. *PloS one* **12**, e0176934 (2017).
<https://doi.org/10.1371/journal.pone.0176934>

2. Did the authors measure the actual warming rate achieved in the tissue or rely on the modelling? The information in this regard given in Supplementary Note 2 is quite relevant and I think it would benefit from being included in the main text.

Response: We have measured the actual warming rate at the cartilage surface for both nanowarming and convection warming methods. These experimental data are included in the updated Fig. 5f-5g of the revised manuscript. Please see the updated Fig. 5 below. The green line and light green shaded ribbon in Fig. 5f showed the average temperature value collected from the experimental measurements at the cartilage surface and its standard deviation, respectively; experimental heating rates were updated in Fig. 5g. As suggested by the reviewer, we have moved the contents in Supplementary Note 2 in the original manuscript to the results and methods section of the revised manuscript (Line 224-242 and Line 590-594). We thank the reviewer for this suggestion.

Minor corrections:

1. Most uses of “rewarmed” throughout the work should be changed to “warmed”

Response: As suggested by the reviewer, we have made all the changes in the revised manuscript.

2. What is the advantage of this method over the microwave warming stated in line 94, stating this explicitly would help the impactfulness of the work.

Response: Heating tissues with microwave warming could generate hot spots inside the tissue, causing tissue damage. As suggested by both reviewers, we have added a paragraph to explicitly explain other volumetric warming methods, including laser warming and microwave warming, and their limitations in the introduction of the revised manuscript (Line 89-100).

3. Do the authors have a minimum warming rate in mind for those tissues cryopreserved in the high concentration CPA? Would future work to increase warming rate help or are the rates achieved here sufficient (with damage caused by other parts of the cryopreservation process)?

Response: Our study found that the surface layer of the cartilage was well preserved, indicating the warming rates at the cartilage surface might be sufficient. Therefore, a warming rate of ~ 30 °C/min (the warming rate measured at the cartilage surface) might be good enough to preserve cartilage with VS83. However, we want to note that the critical warming rate depends on many factors, such as the cooling rates ¹. Future studies are needed to quantify the critical warming rate of VS83 for cartilage cryopreservation.

Our results also showed a depth-dependent preservation manner with better cartilage preservation at the cartilage surface than the middle and deep zone of the cartilage, which may be associated with depth-dependent warming rates. Improving the warming rate at the deep region of the cartilage might lead to better full-thickness cartilage preservation. This leads us to propose the injection method at the end of the manuscript as a strategy to achieve this purpose. Our future work will aim to optimize the warming rate in the middle and deep regions of the cartilage to improve cartilage preservation.

References:

1. Hopkins, J. B., Badeau, R., Warkentin, M. & Thorne, R. E. Effect of common cryoprotectants on critical warming rates and ice formation in aqueous solutions. *Cryobiology* **65**, 169-178 (2012). <https://doi.org:10.1016/j.cryobiol.2012.05.010>

4. Line 382, remove word “in”

Response: We have changed this in the revised manuscript.

5. Line 430. I think “stripes” should read “strips”

Response: We thank the reviewer for the correction. We have changed this in the revised manuscript.

Response to Reviewer #2:

General comment:

The manuscript is very well written and clearly outlines the problem at hand. It explains the need for OCA preservation and its benefits to medicine. The authors have done a great job at outlining the methods for all the biological survival assays and mechanical strength testing experiments.

Response: We thank the reviewer’s appreciation of our work. We have provided responses to each of the reviewer’s specific comments below.

Specific comments:

1. The manuscript does not discuss or show the successful vitrification of the OCA in the current study. It would be beneficial to show an image and cooling rate profile of a vitrified OCA.

Response: We thank the reviewer for raising this question. The vitrification of OCA is successful in our study. As suggested by the reviewer, we have included an image of the vitrified OCA in VS83 and the cooling profiles measured at the cartilage surface of the OCA in the Supplementary Fig. 1 of the revised manuscript (please see the updated figure below).

2. In Fig 2b, it seems like the nanowarmed OCA has a higher green fluorescence signal in the superficial zone compared to the fresh controls. Is there a reason for this to happen? Although the authors mention that nanoparticles do not penetrate in the OCA, it would be a good exercise to perform the assay on a control group wherein all the nanoparticle and CPA solution loading steps and unloading steps are performed without vitrification and rewarming to ensure there is no background signal from the nanoparticles.

Response: The dimmer fluorescence at the superficial zone of the fresh control group is because the sample is not perfectly flat against the glass when performing the imaging. The sample was manually dissected into thin slices for imaging, and it was challenging to make sure the cut surface was perfectly flat across the sample. However, we can still see the cells in the superficial zone of the fresh cartilage in green color, meaning they are alive and not dead.

Our MRI data and visual inspection results showed that the nanoparticles do not penetrate in the OCA (Fig. 5 and Supplementary Fig. 6 in the manuscript). The nanoparticles surrounding the OCA were washed away during the CPA unloading steps. As suggested by the reviewer, we have performed live/dead staining with fresh samples only going through the CPA loading, the nanoparticle solution immersion, and CPA unloading steps but without vitrification and rewarming. The results are shown below. We can see that the chondrocytes in this loading/unloading only group are mostly alive throughout the cartilage thickness. We don't think there are any background signals from the nanoparticles.

Data presented in the main Fig. 2b of the manuscript

3. The authors use a methyl butane bath at -135°C than the more commonly used liquid nitrogen bath at -196°C . What are the benefits of the current thermal protocol?

Response: Cooling the sample by directly placing it in a liquid nitrogen bath (-196°C) can generate a significant thermal gradient across large-sized samples, which might lead to cracks as reported in literature ¹. Therefore, we decided to use a methyl butane bath at -135°C to avoid such detrimental effects. Besides, the VS83 solution requires a relatively slow critical cooling rate due to its high concentration, which permits us to use a methyl butane bath for the cooling. Our results have also shown that the vitrification is successful with our protocol (Supplementary Fig. 1 in the revised manuscript). In addition, using a storage freezer instead of liquid nitrogen to preserve or store samples can potentially reduce the cost of both long-term storage and distribution, lower the risk of microbial contamination associated with the use of liquid nitrogen, and is relatively easy to operate given the safety concerns with liquid nitrogen ².

References:

1. Etheridge, M. L. *et al.* RF heating of magnetic nanoparticles improves the thawing of cryopreserved biomaterials. *TECHNOLOGY* **02**, 229-242 (2014). <https://doi.org/10.1142/s2339547814500204>
2. Brockbank, K. G. M., Chen, Z., Greene, E. D. & Campbell, L. H. in *Cryopreservation and Freeze-Drying Protocols* (eds Willem F. Wolkers & Harriette Oldenhof) 399-421 (Springer New York, 2015). https://doi.org/10.1007/978-1-4939-2193-5_20

4. The concentration of mIONPs used in the current study is 2 mg/ml. Where studies performed to optimize the concentration?

Response: 2 mg/ml is the maximal concentration of mIONPs that can be made with VS83 solutions. VS83 has a very high cryoprotectant concentration (12.6 M). The nanoparticles used in

this study are water-based ferrofluids from the provider. We can only mix a limited amount of nanoparticles into the VS83 solution without diluting the concentration of the cryoprotectants. As the heating rate increases with increasing mIONPs concentration shown in the Supplementary Fig. 5h of the revised manuscript, we used the maximal concentration of mIONPs in this study to achieve the ultimate heating rate of the system. Our results also showed that the cartilage surface layer was well preserved with our current method, indicating the heating rate achieved with 2 mg/ml might be sufficient for cartilage preservation. Furthermore, as indicated by our experimental and modeling data, the observed depth-dependent cartilage preservation manner mainly results from the absence of mIONPs within the OCA rather than insufficient mIONP concentration in the CPA solution. Delivery of the mIONPs to the bone side via injection, as proposed in the manuscript, could possibly solve this issue.

5. Since the manuscript mentions the nanoparticles did not penetrate in the graft, did the authors look into other volumetric heating methods instead? How would nanowarming be a more optimal approach in this case compared to other volumetric heating methods?

Response: We have considered volumetric heating methods other than nanowarming. Microwave warming and laser warming are two of the main volumetric warming methods. However, both methods have limitations. Microwave warming has the major issue of hot spot generation (i.e., heterogeneous heating). Samples might be overheated in certain regions but not efficiently heated at other locations. Laser warming is mainly used for microliter small sample cryopreservation, such as cells or embryos. Also, laser warming relies on the laser absorbing particles, which could have similar issues as we have here. Future developments in these methods might allow us to use them for cartilage cryopreservation. However, at this moment, we think nanowarming is a better choice, particularly when the injection strategy is used for nanoparticle delivery to the bone side to further improve the warming rate at the deep region of the cartilage.

In this regard, we have included a paragraph to compare nanowarming method with other volumetric methods in the introduction of the revised manuscript (Line 89-100).

6. The authors suggest delivery of mIONPs to the bone side of OCA for rapid rewarming. One challenge to this method is the mIONPs washout after rewarming. Have the authors developed a strategy for successful washout of mIONPs?

Response: We could possibly wash out some of the nanoparticles during the CPA unloading step. Also, it is good to note that a bone lavage process is recommended to remove the bone marrow in the clinical OCA implantation surgery to minimize the immune response and improve bone integration¹. This lavage process could help wash the nanoparticles out as well. We don't expect the nanoparticles will be thoroughly washed out from the bone side. However, as the nanoparticles are biocompatible and can degrade over time in the body, a low concentration of residual nanoparticles might not pose a devastating impact on the health of the OCA recipient; this biocompatibility is supported by recent publications on the rat heart² and rat kidney³ nanowarming applications. Our future studies will investigate the optimized wash-out protocol for the injection method.

References:

1. Sun, Y. *et al.* Pulsed lavage cleansing of osteochondral grafts depends on lavage duration, flow intensity, and graft storage condition. *PloS one* **12**, e0176934 (2017). <https://doi.org:10.1371/journal.pone.0176934>

2. Gao, Z. *et al.* Vitrification and Rewarming of Magnetic Nanoparticle-Loaded Rat Hearts. *Advanced materials technologies* **n/a**, 2100873 (2021). <https://doi.org:https://doi.org/10.1002/admt.202100873>

3. Sharma, A. *et al.* Vitrification and Nanowarming of Kidneys. *Advanced science (Weinheim, Baden-Wuerttemberg, Germany)*, e2101691 (2021). <https://doi.org:10.1002/advs.202101691>

7. The concentration of mIONPs was assumed to be half or equal to that of the surrounding CPA solution in the bone side OCA for the nanoparticle delivery method. It would benefit the reader to know the reasoning behind assuming these concentrations.

Response: We thank the reviewer for this suggestion. These concentrations were selected to demonstrate that the warming rate at the deep region of the cartilage is increasing as the nanoparticle concentration increases on the bone side. These results could be presented as a guide for future studies to optimize the nanoparticle loading protocol. As suggested by the reviewer, we have added our intention regarding the concentration selection in the main text of the revised manuscript (Line 280-283).

8. The authors mention another study which was performed on a smaller cartilage by depleting the surrounding CPA solution. It would be interesting to see the comparison between the two methods for the same size of OCA, especially since the mIONPs do not penetrate the OCA. Would nanowarming also be superior in this case?

Response: We thank the reviewer for this question. Both the literature method and our method are trying to improve the heating efficiency and therefore the cryopreservation outcome. However, we believe directly placing a vitrified OCA without surrounding CPA into a water bath could potentially cause a great temperature gradient across the sample, which might cause tissue damage, especially when the sample size gets bigger. We also believe nanowarming is still a favorable approach as we could inject the nanoparticles into the bone region to ramp up the heating rate at the deep region of the cartilage, while the CPA depletion method doesn't have this flexibility. Furthermore, as stated in the discussion section, our approach could allow easier sample storage. We have included these discussions in the revised manuscript (Line 316-319). We don't intend to compare these two methods in this manuscript, but our future studies can investigate the differences between these two methods.

9. Fig 5b/6c: The contrast/LUT of the MRI scans can be calibrated and optimized for viewing of nanoparticles. It is difficult to observe on a grayscale. It would also benefit the reads to see one image with CPA loaded nanoparticles to be able to see the difference with and without nanoparticles.

Response: We thank the reviewer for this suggestion. We have added the MRI image of different concentrations of nanoparticles in 1% agarose gel with the same MRI scanning protocols used with the OCA samples to give an MRI image contrast reference. The updated images are presented in Supplementary Fig. 6a-6b of the revised manuscript (please see the updated figure below).

10. Fig 5f-e: Due to the absence of mIONPs in the OCA, it would be interesting to see the temperature profile at the center of the bone in the computational model. Due to thermal inertia do the faster rewarming rates closer to the outer surfaces actually translate to the center of the bone? How much is the rewarming rate difference at the center using the two rewarming methods?

Response: We have extracted the data from our model, and the temperature profile and heating rate at the bone center can be seen below:

The warming rate at the bone center is much lower than at the cartilage layer as the temperature rise in the bone center relies on heat transfer from the outer surface of the OCA. However, the heating rate in the nanowarming group is still 2-3 times higher than that in the convection

warming group. We have realized the warming issue at the bone side. This is also part of the reason why we proposed to inject nanoparticles inside the bone to augment the warming rate at the bone side. Additionally, to our best knowledge, there are not many studies on bone warming for OCA cryopreservation. Our future studies will include the warming on the bone side.

11. Have the modeling results been validated against the experimental results specifically with the OCA present? One way to do this would be to place thermocouples in the experimental set up and computational model at the same location and compute the difference.

Response: We thank the reviewer for the question and suggestion. As suggested by the reviewer, we have performed temperature measurements at the cartilage surface for both nanowarming and convection warming with OCAs and compared them to the modeling values at the same location. The results are included in the updated Fig. 5f-5g of the revised manuscript. Please see the updated figure below. The green line and light green shaded ribbon in Fig. 5f showed the average temperature value collected from the experimental measurements at the cartilage surface and its standard deviation, respectively; experimental heating rates were updated in Fig. 5g. Overall, our experimental results agree with the modeling findings: the heating rates were significantly higher in the nanowarming group than in the convection warming group. The modeling heating rate was slightly higher than the experimental heating rate value, which might be related to the use of temperature-independent thermal properties of the CPA solution in our current model. Our future work will investigate the CPA thermal properties in detail, therefore having all the input parameters for a more accurate model for cartilage nanowarming.

12. Line 537: There 'is' very limited VS83 thermal data available in the literature.

Response: We apologize for this typo error. We have corrected it in the updated manuscript.

13. Line 254-257: This statement is not entirely true due to the non-uniform temperature profile observed in the OCA. Thermal stresses can still be formed due to non-uniform heating. The key to reduce thermal stresses is uniformity in rewarming rates, which might still be a challenge due to the absence of nanoparticles in the OCA.

Response: We agree with the reviewer's comments. We have removed the statements about thermal stress (Line 291). We will try to further optimize our protocol to achieve more uniform warming across the OCA and investigate the thermal stress impacts in our future study.

Reviewers' comments:

Reviewer #1 (Remarks to the Author):

The authors' have answered all of the points raised throughly. I recommend publishing the work in its current form.

Reviewer #2(Remarks to the Author):

Overall comment:

The manuscript is substantially improved after the revision.

Specific comments:

In figure 5(f), the modeling and experimental temperature profiles have significant temperature differences which would only increase with increasing temperatures all the way to 0C. This would suggest the computational model is not an accurate representation of the experimental set up. This could possibly be due to (a) lower specific heat capacity value for VS55 used, (b) use of adiabatic boundary condition instead of free boundary condition during nanowarming, and/or (c) inaccurate properties for the falcon tube material used for computational analysis. Please refer the references mentioned below for the temperature dependent and updated thermal properties for VS55.

If the differences still persist after using updated properties, comment about that in the manuscript to acknowledge the discrepancies between the computational and experimental observations.

References:

- (1)Joshi, P., Ehrlich, L. E., Gao, Z., Bischof, J. C., & Rabin, Y. (2022). Thermal Analyses of Nanowarming-Assisted Recovery of the Heart From Cryopreservation by Vitrification. *Journal of Heat Transfer*, 144, 031202 1-11. <https://doi.org/10.1115/1.4053105>
- (2) Etheridge, M. L., Xu, Y., Rott, L., Choi, J., Glasmacher, B., & Bischof, J. C. (2014). RF heating of magnetic nanoparticles improves the thawing of cryopreserved biomaterials. *Technology*, 02(03), 229–242. <https://doi.org/10.1142/s2339547814500204>

Re: COMMSBIO-22-2077A

Nanowarming and ice-free cryopreservation of large sized, intact articular cartilage

Response to Reviewer #1:

The authors' have answered all of the points raised throughly. I recommend publishing the work in its current form.

Response: We thank the reviewer for their time, suggestions, and critical comments, which help us improve the quality of our manuscript.

Response to Reviewer #2:

Overall comment:

The manuscript is substantially improved after the revision.

Response: We thank the reviewer for their time and appreciation of our efforts in addressing their previous comments. We have responded to the reviewer's specific comments below.

Specific comments:

In figure 5(f), the modeling and experimental temperature profiles have significant temperature differences which would only increase with increasing temperatures all the way to 0C. This would suggest the computational model is not an accurate representation of the experimental set up. This could possibly be due to (a) lower specific heat capacity value for VS55 used, (b) use of adiabatic boundary condition instead of free boundary condition during nanowarming, and/or (c) inaccurate properties for the falcon tube material used for computational analysis. Please refer the references mentioned below for the temperature dependent and updated thermal properties for VS55.

If the differences still persist after using updated properties, comment about that in the manuscript to acknowledge the discrepancies between the computational and experimental observations.

References:

(1)Joshi, P., Ehrlich, L. E., Gao, Z., Bischof, J. C., & Rabin, Y. (2022). Thermal Analyses of Nanowarming-Assisted Recovery of the Heart From Cryopreservation by Vitrification. *Journal of Heat Transfer*, 144, 031202 1-11. <https://doi.org/10.1115/1.4053105>

(2) Etheridge, M. L., Xu, Y., Rott, L., Choi, J., Glasmacher, B., & Bischof, J. C. (2014). RF heating of magnetic nanoparticles improves the thawing of cryopreserved biomaterials. *Technology*, 02(03), 229–242. <https://doi.org/10.1142/s2339547814500204>

Response: We thank the reviewer for the comments. We agree with the reviewer that the modeling temperature profile shown in Fig. 5f differs from the experimental measurements at later times. We also agree with the reviewer that the discrepancies between modeling results and experimental data might result from using temperature-independent CPA thermal properties in our model. We appreciate the reviewer's suggestions and the references for updated CPA thermal properties. We acknowledge that the CPA thermal properties are temperature-dependent. In our model, we used the average CPA thermal parameters, like the one reported in literature ¹, as a simplified model to reveal the possible causes of our experimental depth-dependent preservation observations and hint the potential means to improve the outcome further. Future studies are needed to build a comprehensive model that more closely represents the actual nanowarming with the OCA sample.

As recommended by the reviewer, we have performed some new simulations with the updated CPA thermal properties to demonstrate whether the use of temperature dependent CPA thermal properties affects the OCA nanowarming modeling results. Along with the VS55 thermal properties suggested by the reviewer ^{2,3}, we also find another set of VS55 thermal data ^{4,5}. We included both literature VS55 thermal property values in the following simulations. The same model geometry used in Fig. 5f was applied, and the heat generation rate was set the same, $Q=2.956 \times 10^6 \text{ W/m}^3$.

Fig. a showed the two sets value of VS55 specific heat capacity (C_p) reported in the literature: the red line indicated the one reported in the references suggested by the reviewer^{2,3} (marked as $C_p(T)1$), and the purple dots was the one reported in other literature^{4,5} (marked as $C_p(T)2$). The black line indicated the constant C_p value ($C_p=2100$) used in our manuscript, which is regarded as the average C_p value across the temperature range investigated in our study¹. **Fig. b** showed the value of VS55 thermal conductivity reported in the literature ($k=0.3$ and $k(T)$ were labeled for the temperature independent^{1,3,4,6} and dependent^{2,7} thermal conductivity, respectively).

In **Fig. c**, we simulated the OCA nanowarming with three sets of C_p values shown in **Fig. a**. We also compared the modeling results with adiabatic and free boundary conditions. In the case of free boundary conditions, the heat transfer coefficient (h) was set of $15 \text{ W/m}^2\cdot\text{K}$ as reported in the literature^{2,4}. Our results demonstrated the C_p value affects the temperature curve. The temperature rise was fastest in the model with $C_p(T)1$ and slowest in the model with $C_p(T)2$. The results are expected as the overall C_p value is the smallest for $C_p(T)1$ and largest in $C_p(T)2$ in the temperature range studied in our current experimental nanowarming protocol. Our modeling results using constant C_p value are within the range of the modeling results with the two temperature-dependent C_p values. In addition, our results showed that the boundary conditions have minimal impact on the temperature profile at the cartilage surface. The results are anticipated as the heat transfer between the sample, and the atmosphere is very limited compared to the heat generated by nanowarming.

In **Fig. d**, we compared the modeling results with a temperature-independent and dependent thermal conductivity value shown in **Fig. b** for all three sets of C_p values shown in **Fig. a**. Free boundary conditions were used for all the simulations in this subfigure. Adiabatic boundary conditions were not simulated, as there would be no temperature gradient within the sample. Therefore, the value of thermal conductivity will not change the temperature profile. Our results demonstrated that using either temperature-dependent or independent conductivity values won't change the temperature profile very much at the cartilage surface. These results are also reasonable as the heat transfer between the sample, and the atmosphere is very limited.

In our model, we did not include the falcon tube. Therefore, we did not add the simulation results with the falcon tube thermal properties suggested by the reviewer here. We believe the impact of the additional falcon tube layer on the temperature profile at the cartilage surface would be minor as the heat transfer between the sample and the atmosphere is very limited compared to the heat generated through nanowarming (as shown in Fig. c and d).

Overall, the modeling results showed that using different CPA specific heat capacity values can impact the temperature profile at the cartilage surface in nanowarming. These results demonstrate the importance of measuring the thermal properties of different CPA solutions for accurately modeling the nanowarming process. However, as shown here, multiple sets of C_p values are reported in the literature. Additionally, although we have shown similar thermal behavior between VS55 and VS83 during the cooling and warming process, there could still be some differences in their thermal properties. Future investigation of VS83 thermal properties is still needed. Furthermore, the heat generation rate due to nanowarming has also been reported as temperature dependent^{2,4,7}, further complicating the model. The CPA-loaded OCA thermal

properties were also not determined. Following prior studies^{2,4,7}, they were assumed to be the same as the pure CPA thermal properties in our model, which might cause some deviations of our modeling results from the experimental measurements. Our future work will investigate these missing pieces and build a more comprehensive model for the nanowarming with the cartilage sample. At this point, we would like to keep our results in their current form and acknowledge the discrepancies between the modeling and experimental results, as suggested by the reviewer. We have included a new paragraph in the Discussion section (Line 404-417).

References:

- 1 Manuchehrabadi, N. *et al.* Improved tissue cryopreservation using inductive heating of magnetic nanoparticles. *Science translational medicine* **9**, eaah4586, doi:10.1126/scitranslmed.aah4586 (2017).
- 2 Joshi, P., Ehrlich, L. E., Gao, Z., Bischof, J. C. & Rabin, Y. Thermal Analyses of Nanowarming-Assisted Recovery of the Heart From Cryopreservation by Vitrification. *J Heat Transfer* **144**, 031202, doi:10.1115/1.4053105 (2022).
- 3 Etheridge, M. L. *et al.* RF heating of magnetic nanoparticles improves the thawing of cryopreserved biomaterials. *TECHNOLOGY* **02**, 229-242, doi:10.1142/s2339547814500204 (2014).
- 4 Sharma, A. *et al.* Vitrification and Nanowarming of Kidneys. *Advanced science (Weinheim, Baden-Wuerttemberg, Germany)*, e2101691, doi:10.1002/advs.202101691 (2021).
- 5 Phatak, S., Natesan, H., Choi, J., Brockbank, K. G. M. & Bischof, J. C. Measurement of Specific Heat and Crystallization in VS55, DP6, and M22 Cryoprotectant Systems With and Without Sucrose. *Biopreservation and biobanking* **16**, 270-277, doi:10.1089/bio.2018.0006 (2018).
- 6 Eisenberg, D. P., Bischof, J. C. & Rabin, Y. Thermomechanical Stress in Cryopreservation Via Vitrification With Nanoparticle Heating as a Stress-Moderating Effect. *Journal of Biomechanical Engineering* **138**, doi:10.1115/1.4032053 (2015).
- 7 Gao, Z. *et al.* Vitrification and Rewarming of Magnetic Nanoparticle-Loaded Rat Hearts. *Advanced materials technologies* **n/a**, 2100873, doi:https://doi.org/10.1002/admt.202100873 (2021).

REVIEWERS' COMMENTS:

Reviewer #4 (Remarks to the Author):

The authors have addressed all the comments posed by the reviewers. The manuscript has greatly improved after the revisions.

Re: COMMSBIO-22-2077B

Nanowarming and ice-free cryopreservation of large sized, intact porcine articular cartilage

Response to Reviewer #4:

The authors have addressed all the comments posed by the reviewers. The manuscript has greatly improved after the revisions.

Response: We thank the reviewer for their time and their acknowledgement of our efforts in addressing their previous questions.